# Functional comparison of MERS-coronavirus lineages reveals increased replicative fitness of the recombinant lineage 5

Simon Schroeder[1], Christin Mache[2], Hannah Kleine-Weber[3], Victor M. Corman [1,4], Doreen Muth[1], Anja Richter[1], Diana Fatykhova[5], Ziad A. Memish[6,7,8], Megan L. Stanifer [9], Steeve Boulant[10,11], Mitra Gultom[12,13,14,15], Ronald Dijkman [12,13,14], Stephan Eggeling[16], Andreas Hocke[5], Stefan Hippenstiel [5], Volker Thiel [12,13], Stefan Pöhlmann [3], Thorsten Wolff [2], Marcel A. Müller [1,4,17] & Christian Drosten [1,4✉]

Middle East respiratory syndrome coronavirus (MERS-CoV) is enzootic in dromedary camels across the Middle East and Africa. Virus-induced pneumonia in humans results from animal contact, with a potential for limited onward transmission. Phenotypic changes have been suspected after a novel recombinant clade (lineage 5) caused large nosocomial outbreaks in Saudi Arabia and South Korea in 2016. However, there has been no functional assessment. Here we perform a comprehensive in vitro and ex vivo comparison of viruses from parental and recombinant virus lineages (lineage 3, n = 7; lineage 4, n = 8; lineage 5, n = 9 viruses) from Saudi Arabia, isolated immediately before and after the shift toward lineage 5. Replication of lineage 5 viruses is significantly increased. Transcriptional profiling finds reduced induction of immune genes *IFNB1*, *CCL5*, and *IFNL1* in lung cells infected with lineage 5 strains. Phenotypic differences may be determined by IFN antagonism based on experiments using IFN receptor knock out and signaling inhibition. Additionally, lineage 5 is more resilient against IFN pre-treatment of Calu-3 cells (ca. 10-fold difference in replication). This phenotypic change associated with lineage 5 has remained undiscovered by viral sequence surveillance, but may be a relevant indicator of pandemic potential.

[1] Institute of Virology, Charité-Universitätsmedizin Berlin, Berlin, Germany. [2] Unit 17, Influenza and other Respiratory Viruses, Robert Koch Institut, Berlin, Germany. [3] Infection Biology Unit, German Primate Center – Leibniz Institute for Primate Research, Göttingen, Germany. [4] German Centre for Infection Research (DZIF), Berlin, Germany. [5] Dept. of Infectious and Respiratory Diseases, Charité-Universitätsmedizin Berlin, Freie Universität Berlin, Humboldt-Universität zu Berlin, and Berlin Institute of Health, Berlin, Germany. [6] Research and Innovation Department, King Saud Medical City, Ministry of Health, Riyadh, Saudi Arabia. [7] College of Medicine, Alfaisal University, Riyadh, Kingdom of Saudi Arabia. [8] Hubert Department of Global Health, Rollins School of Public Health, Emory University, Atlanta, GA, USA. [9] Department of Infectious Diseases, Molecular Virology, Heidelberg University Hospital, Heidelberg, Germany. [10] Research Group "Cellular polarity and viral infection", German Cancer Research Center (DKFZ), Heidelberg, Germany. [11] Department of Infectious Diseases, Virology, Heidelberg University, Heidelberg, Germany. [12] Institute of Virology and Immunology (IVI), Bern, Switzerland. [13] Department of Infectious Diseases and Pathobiology, Vetsuisse Faculty, University of Bern, Bern, Switzerland. [14] Institute for Infectious Diseases, University of Bern, Bern, Switzerland. [15] Graduate School for Biomedical Science, University of Bern, Bern, Switzerland. [16] Department of Thoracic Surgery, Vivantes Clinics Neukölln, Berlin, Germany. [17] Martsinovsky Institute of Medical Parasitology, Tropical and Vector Borne Diseases, Sechenov University, Moscow, Russia. ✉email: christian.drosten@charite.de

The pandemic spread of severe acute respiratory coronavirus 2 (SARS-CoV-2) demonstrates the importance of monitoring emerging coronaviruses (CoV). Middle East respiratory syndrome coronavirus (MERS-CoV) is the causative agent of severe viral pneumonia in humans[1]. MERS-CoV emerged in 2012 and the majority of currently more than 2494 notified cases, including 858 deaths, have occurred on the Arabian Peninsula. Travel-associated cases were diagnosed in 27 countries and sparked secondary case clusters in some. The largest MERS-CoV outbreak outside the Arabian Peninsula occurred in South Korea in 2015, involving 186 cases and 36 deaths (WHO, 2020).

MERS-CoV is acquired as a zoonotic infection from dromedary camels[2–5]. Spill-over from dromedaries to humans can lead to local outbreaks with limited human-to-human transmission[6,7]. Healthcare facilities can experience protracted outbreaks with severe infections in comorbid patients[8]. Behavioral factors like family patient care may accelerate outbreaks[9–11].

MERS-CoV phylogeny currently comprises three major clades, provisionally named clades A, B, and C[12–14]. Whereas clades A and C contain extinct strains and strains not circulating in the Arabian Peninsula, clade B strains currently infect humans and dromedary camels in this area. Clade B is subdivided into six phylogenetic lineages. Presumed recombination between lineage 3 and 4 resulted in the formation of a circulating recombinant lineage (lineage 5, also termed NRC for novel recombinant clade) during or before the year 2014 in dromedary camels[14–17].

Limited diagnostic testing, limited surveillance, and improved hospital infection control may have caused an apparent decline of notified cases after 2015[18]. Only based on more recent studies, it appears that lineage 5 has essentially replaced all other endemic strains since 2015[19,20] (Fig. 1). Because clade B lineages used to co-circulate widely before 2015, the novel dominance of a single viral lineage deserves clarification. Any change of phenotype might indicate alterations in the already existing potential for human-to-human transmission. Changes of phenotype in association with lineage 5 emergence have been suspected, but phenotypic studies of MERS-CoV strains are generally limited. Mutations in the spike protein positions I529T and D510G observed during the outbreak in Korea were suggested to have contributed to antibody evasion[21]. However, these polymorphisms evolved during and not prior to the Korean MERS-CoV outbreak and hence cannot explain the dominance of lineage 5 since 2015[22]. Earlier studies looking into functional differences between MERS-CoV clades other than lineage 5 found little evidence for phenotypic differences[12,23–25]. One complicating

feature of MERS-CoV is that the infection phenotype seen in humans is difficult to reflect in small animal models.

In this work, we investigate phenotypic traits in phylogenetically distinct MERS-CoV lineages including the recombinant lineage 5, using different cell-, epithelial-, and ex vivo human lung models. As MERS-CoV is known to act against the induction of cytokines and overcome the effects of antiviral genes[26–30], we investigate differences in immune gene activation and suppression of viral replication in response to interferon (IFN) signaling. We find lineage 5 viruses replicate more efficiently, show decreased antiviral IRF3- and NFkB-dependent signaling, and are less susceptible to the IFN response.

## Results

**Phylogeny and recombination analysis of MERS-CoV isolates obtained from Saudi Arabian patients.** To facilitate a phenotypic comparison of MERS-CoV strains that circulated before and after the 2014 recombination event, we generated 23 virus isolates from patient samples obtained during outbreaks in Saudi Arabia between 2014 and 2015 and one virus isolate obtained from a dromedary camel farm in Dubai in 2017 using a previously established MERS-CoV isolation protocol[25]. Early passage isolates were grown to high concentration, physically purified, and characterized by deep sequencing, real-time RT-PCR, and plaque quantification. No plaque purification was performed to avoid the introduction of selection bottlenecks. To further avoid cell culture-derived selection biases, all quantitative experiments were done on different representative isolates per viral clade with subsequent pooling of results.

For phylogenetic classification, we sequenced all isolates and inferred trees based on whole-genome sequences (Supplementary Fig. 1). All studied isolates cluster with either lineage 3, lineage 4, or lineage 5. As expected, lineage 5 branches from lineage 3 in a tree based on a full genome alignment[14,15,17]. Among the present isolates, seven belong to lineage 3 and originate from a MERS-CoV outbreak in Riyadh in 2014[31]. Eight belong to lineage 4 and originate from a 2014 outbreak in Jeddah[24]. Another nine isolates belong to lineage 5. Eight of these stem from patients treated in Riyadh between September and November 2015, one (D896 2017) from a dromedary camel farm in Dubai sampled in 2017. All show the typical pattern of topological incongruence when inferring trees from alignments that cover different genome portions relative to the two recombination breakpoints.

**Replication of MERS-CoV lineages in cell culture.** To compare viral replication we performed multicycle infection experiments in Vero B4 monkey kidney cells that are widely used for diagnostic isolation of MERS-CoV. We observed no significant growth differences between isolates (Fig. 2a). Because Vero cells are known to be deficient in IFN-I induction, we also performed multicycle infection experiments in the human lung- and colon carcinoma cell lines Calu-3 and Caco-2, respectively (Fig. 2b, c). These experiments detected significantly increased replication levels for all tested isolates pertaining to lineage 5. Growth was enhanced over that of the EMC reference strain, but also over that of all tested isolates pertaining to parental lineages 3 and 4 (Fig. 2a–c). Two representative isolates per lineage were tested in these experiments initially. In order to minimize any influence of possible inter-isolate phenotypic variability, we used an extended range of virus isolates for the experiment in Calu-3 cells (Fig. 2b). In each of four independent experiments, we used different sets of two viral isolates per lineage, resulting in the representation of each lineage by eight different viral isolates (refer to Supplementary Table 1 for virus isolates used in each experimental repetition). In addition, we infected Calu-3 cells with a lineage 5

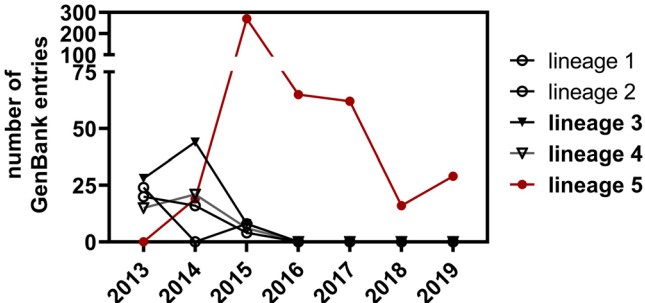

**Fig. 1 MERS-CoV sequences uploaded to NCBI GenBank between 2013 and 2020.** All available GenBank entries for MERS-CoV with a sequence length ≥28,000 nucleotides (n = 537) nucleotides were included, as well as concatemerized sequences with sequence length ≥5,000 nucleotides[19] n = 24. The collection date provided in GenBank was used for assigning a year to each sequence. No sequences were found with a collection date later than 2019.

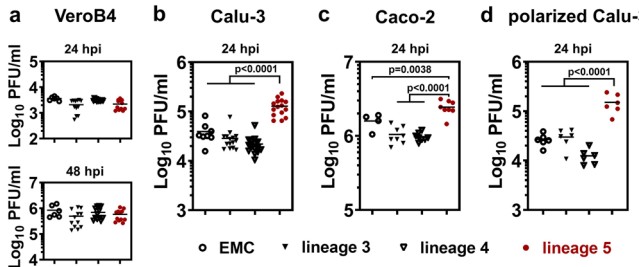

**Fig. 2 Enhanced replication of lineage 5 MERS-CoV isolates on cultured cells.** Cells were infected at MOI = 0.002 and virus progeny in cell culture supernatants was quantified by plaque assays. **a** Vero B4 cells, 24 and 48 hpi; **b** Calu-3 lung cells, 24 hpi; **c** Caco-2 colon cells, 24 hpi; **d** Calu-3 lung cells grown as air-liquid interface culture, 24 hpi. All infections were performed with several representative viral isolates per phylogenetic lineage as documented in Supplementary Table 1. Shown are in **a** the results of $n = 2$ independent experiments with $n = 2$ MERS-CoV isolates per lineage and biological triplicates. In **b** the results of $n = 4$ independent experiments overall including all 23 MERS-CoV isolates using biological duplicates. In **c** the results of $n = 2$ independent experiments with $n = 2$ MERS-CoV isolates per lineage and biological duplicates. In **d** the results of $n = 2$ independent experiments with $n = 1$ MERS-CoV isolate per lineage and biological triplicates.; Statistical significance in difference of PFU/ml between groups was determined by one-way ANOVA, followed by multiple two-tailed Student´s $t$-tests between lineage 5 and other lineages. PFU, plaque forming units; hpi, hours post infection.

MERS-CoV isolate (D896) taken in 2017 and found this isolate to exhibit the same replicative advantage as the 2015 lineage 5 isolates (Supplementary Fig. 2). These findings indicate that the increased replication of lineage 5 MERS-CoV may be a shared feature of the whole phylogenetic lineage independent of time or place of virus isolation.

On broad average, all tested isolates pertaining to lineage 5 produced viral titers at least fivefold higher at 24 hpi than lineage 3 and lineage 4 isolates (Fig. 2b, c). Within individual experiments, representatives of lineage 5 reached up to tenfold higher infectious titers than those of lineage 3 and 4. The experiment with multiple replicates and isolates in Fig. 2b yielded an average concentration of 1.23x10e5 PFU/ml for lineage 5 isolates compared to 2.2x10e4 PFU/ml for lineage 3 and 2.4x10e4 PFU/ml for lineage 4. All of these differences were highly significant.

To better represent the conditions of viral replication in respiratory epithelia, we infected polarized Calu-3 cells grown under air-liquid interface (ALI) conditions and found the same relative viral replication phenotypes (Fig. 2d).

**Competitive replicative fitness.** Cell culture growth kinetics can suffer from systematic and random errors, parts of which can be controlled for by competitive replication studies in which viral isolates compete in the same culture dish. A given virus is likely to have superior relative fitness if it can become dominant in a virus population in spite of starting as a minority population in the initial virus seed dose used for infection[32]. To test this, Calu-3 cells were infected with a mixture of a representative virus of lineage 5 and lineage 3 in two different ratios (lineage 3: lineage 5 ratio, 1:1 and 9:1, respectively). The initial seed dose in these cultures was set to 10,000 plaque-forming units (PFU), corresponding to an MOI of 0.04, which enables a short multicycle growth experiment while avoiding stochastic bottleneck effects due to low seed dose. As the total virus amplification from a single round of infection starting from this seed dose is too limited to detect robust differences in replicative fitness, four additional amplification cycles (passages) were performed. In

preliminary experiments in the given cell culture format, we determined that the yield of infectious virus progeny when infecting Calu-3 cells at MOI = 0.04 will be 10e4–10e5 PFU/ml at 24 hpi (Supplementary Fig. 3). We, therefore, diluted passage supernatants to a new seed dose of ca. 5,000 PFU and repeated this process until the completion of a total of five passages. We isolated viral RNA from the initial inoculum (p0) and from the supernatant after five passages (p5) and directly sequenced two different single nucleotide polymorphisms (SNP) sites that were each amplified from the virus population in three separate RT-PCR reactions to control for PCR-based artifacts.

Based on Sanger sequencing we analyzed peak heights to quantify how much of each virus lineage was present at p0 and p5 (Fig. 3a). In both analyzed positions, we found the ratio of virus progeny at p5 to shift in favor of lineage 5 (Fig. 3b).

**Analysis of stages of the viral replication cycle.** Multiple factors might be responsible for enhanced MERS-CoV growth in cell cultures, including improved virus attachment, entry, transcription, replication, infectious particle production, or innate immune counteraction.

**Spike protein gene sequences.** We found no indications for differences in sialic acid-binding and receptor-binding properties in amino acid alignments of viral spike sequences (Supplementary Fig. 4). The N-terminus mediates virus attachment via sialic acid domains. Its sequence is identical in all four lineages and in silico analysis by NetNGlyc and NetOGlyc showed no differences in glycosylation patterns. The receptor-binding domain (RBD) is responsible for binding to the entry receptor, DPP4. The four lineages share the same amino acid sequence in the RBD, with the exception of lineage 3 isolates showing one polymorphism (L411F). However, previous studies of this polymorphism demonstrated unaltered binding affinity and entry efficiency[21].

**Cell entry capacity and receptor binding.** To explore cell entry capacity and receptor binding, we produced expression vectors carrying the spike genes of lineage 3, lineage 4, and lineage 5 strains and incorporated these into vesicular stomatitis virus (VSV)-based pseudotypes[21]. We tested entry capacity in cell lines expressing high and low amounts of TMRPSS2 (high: Calu-3 and Caco-2, Fig. 4a, b; low: Huh7, Fig. 4c) to account for entry capacity with protease-primed and non-primed spike protein[33]. No differences were observed between pseudotypes.

We also expressed the lineage-specific spike proteins in 293 T cells and measured the binding capacity to the MERS-CoV entry receptor, DPP4, by flow cytometry as described in ref. [21]. As a control, we included the previously described variant I529T, known to have reduced binding affinity[21]. Again, we found no significant differences in binding affinities between spike proteins of each lineage (Fig. 4d).

**Full virus entry studies.** To investigate the entire entry process, we performed single cycle (MOI = 1) infection assays in Calu-3 lung cells using virus isolates. We infected all cell cultures at 4 °C to ensure that cell entry is initiated simultaneously by temperature shift to 37 °C. The onset of subgenomic N RNA (sgmRNA N) transcription, as observable before the onset of genome replication was taken as an early indicator of the completion of the entry process. As a control, we simultaneously blocked the entry of MERS-CoV on several levels using inhibitors of clathrin-mediated endocytosis (classical entry pathway), host membrane serine proteases (alternative direct entry pathway), as well as endosomal proteases (downstream endosome fusion). We detected no significant differences in genomic and subgenomic

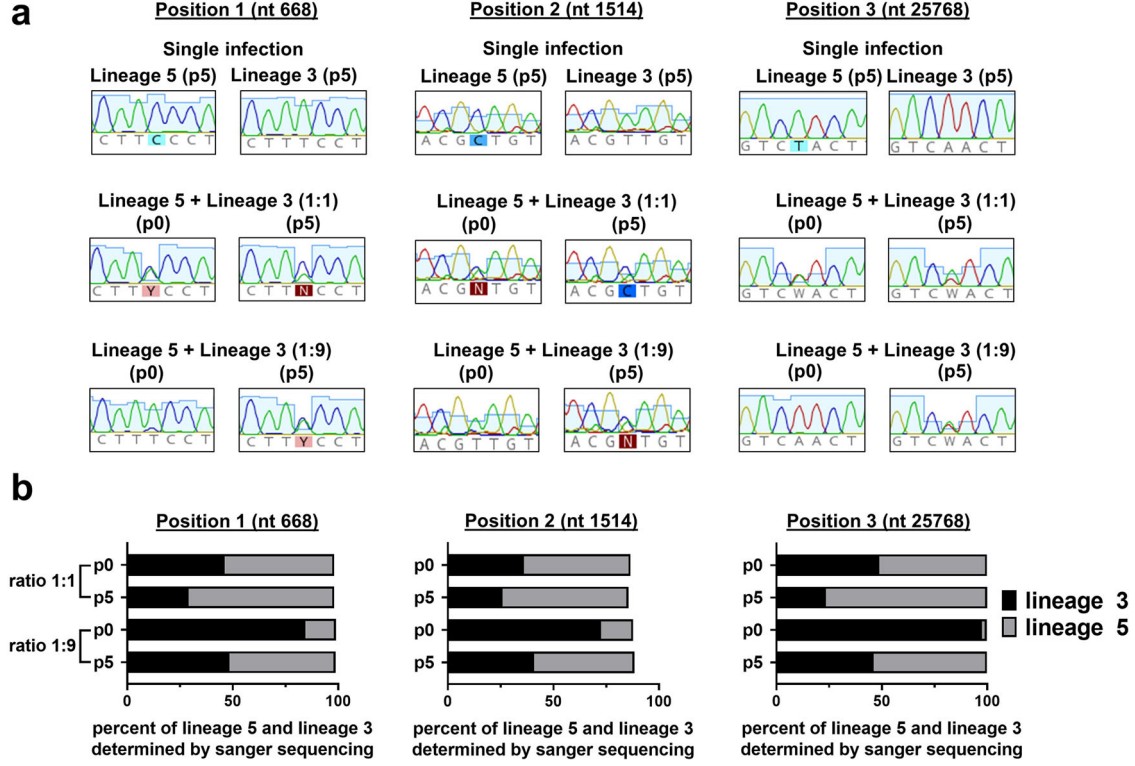

**Fig. 3 Lineage 5 outcompetes a parental virus strain of lineage 3 in an in vitro fitness competition assay.** Calu-3 cells were infected in duplicates with 10,000 PFU containing the indicated ratios of a lineage 3: lineage 5 virus isolate at the time of initial infection (p0). Ca 1,000–10,000 PFU were transferred in five subsequent passages. **a** At p0 and p5, viral RNA was isolated from the inoculum/supernatant and two RT-PCRs covering two SNP that discriminate lineage 3 and lineage 5 were performed. RT-PCR amplicons were subjected to Sanger sequencing to visualize shifts in sequence composition. Data of one representative experiment is shown **b** the average peak heights in sequencing chromatograms at each SNP were analyzed using the ChromatQuantitator server and compared between p0 and p5 populations. nt, nucleotide; p, passage.

RNA quantities among all virus lineages 1 and 4 h post binding (Fig. 5). Chemical blocking of virus entry revealed that all four lineages had entered into the stage of early subgenomic RNA transcription by 4 h post adsorption, without discernible differences in transcription levels.

**Virus neutralization by anti-MERS-CoV-positive human and dromedary camel sera.** Because neutralizing antibodies may act via proteins or domains other than the spike N-terminus and RBD, we tested serum neutralization based on a highly sensitive plaque reduction neutralization assay on live viruses. All virus isolates are neutralized with equal efficiency by human ($N = 2$) and dromedary camel ($N = 3$) sera as summarized and referenced in Supplementary Table 2.

**Early viral replication.** Whereas the above experiments used synchronized infection at high MOI, we also performed multi-cycle, low MOI replication experiments in which the early phase of replication is less masked by input viral RNA and the early onset of transcription may not be discernible from genome replication[26]. Calu-3 and Vero B4 cells were inoculated at 4 °C to ensure that virus particles were attached with equal efficiency to the cellular receptors and the cell entry process started simultaneously. We quantified subgenomic mRNA and genomic RNA transcription and measured PFU/ml in the supernatant to analyze putative differences in virus particle formation and egress (Fig. 6). There was little difference between the amount of N- and genomic transcripts. Virus replication in Calu-3 cells started earlier than in Vero cells, which may be attributed to the availability of

TMPRSS2 providing an additional entry pathway in Calu-3 cells[34]. Already from the beginning of detectable replication, lineage 5 viruses show a higher level of RNA transcription than the parental lineages 3 and 4. Enhanced infectious virus production of lineage 5 was only seen in Calu-3- but not in type I IFN- and TMPRSS2-deficient Vero B4 cells, which corresponds to our previous observations (Fig. 2). As experiments up to this point have largely excluded a role of TMPRSS2-dependent entry, we suspected the phenotypic difference to be linked to cell-intrinsic innate immunity and in particular the type I IFN system.

**Differences in IFN and cytokine induction.** To explore if higher replication is accompanied by higher cytokine induction, we infected Calu-3 cells with two viruses of each phylogenetic lineage in a single-cycle infection for 12 h and analyzed the mRNA expression levels of a set of immune-related genes. We chose IRF3-regulated genes *IFNB1* and *IFNL1*, NFkB-regulated genes *CCL5* and *TNFa*, as well as the IFN-stimulated gene (ISG) *MX1*. To enable single-cycle virus infection, cells were infected at MOI = 2 (Fig. 7a). Also under these conditions, the two lineage 5 isolates used in the experiment replicated to a higher level than the members of lineage 3 and 4. Despite higher replication putatively upregulating ligands for innate immune sensing such as dsRNA, lineage 5 induced significantly lower levels of IFN and *CCL5* mRNA compared to isolates pertaining to lineage 3 and 4 (Fig. 7b). Reduced *IFNB1* and *IFNL1* mRNA expression levels in lineage 5 infected cells might be a result of a more effective viral nucleic acid sequestration from pattern recognition receptors (PRRs) recognition or may reflect an active block of the innate

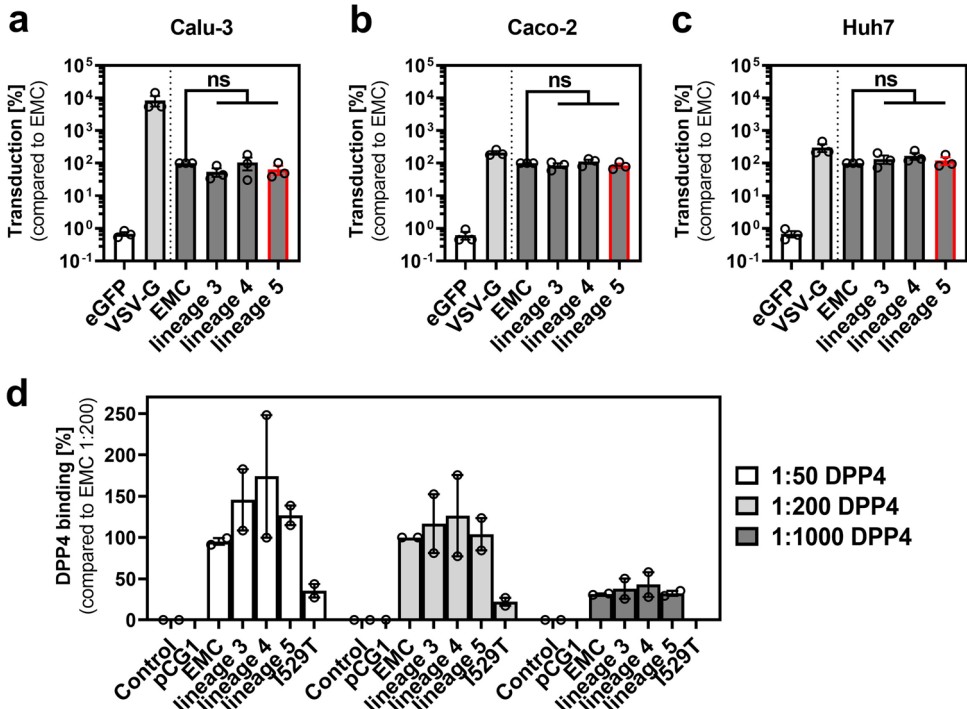

**Fig. 4 Lineage-specific MERS-CoV spike proteins show no difference in host cell entry and DPP4 binding capacity. a–c** rhabdoviral particles harboring MERS-CoV spike proteins of the EMC isolate, lineage 3, lineage 4, and lineage 5, VSV-G (positive control), or eGFP (negative control) were inoculated onto TMPRSS2-expressing Calu-3, Caco-2, and TMRPSS2 negative Huh7 cells. Transduction efficiency was quantified at 18 h post transduction by measuring the activity of virus-encoded luciferase in cell lysates. Transduction mediated by EMC spike protein was set as 100%. The means of $n = 3$ individual experiments performed in biological quadruplicates are shown; error bars indicate SEMs. Statistical significance was analyzed by paired two-tailed Student´s $t$-tests. **d** 293 T cells untransfected or transfected to express MERS-CoV lineage-specific spike proteins or empty expression vector (pCG1) were detached and incubated with human Fc-tagged, soluble DPP4 (solDPP4-Fc), diluted 1:50, 1:200, and 1:1,000, and an Alexa Fluor 488-conjugated antihuman antibody before DPP4 binding was quantified by flow cytometry. For normalization, the binding of solDPP4-Fc (1:200) to EMC spike was set as 100%. For background subtraction of samples incubated with Alexa Fluor 488-conjugated antibodies only (control) was performed for each sample. EMC spike carrying the I529T SNP was included as an internal control, since this SNP has been previously shown to exhibit reduced DPP4 binding[21]. Shown are the normalized data of two independent experiments. Bars indicate mean and SEM. eGFP, enhanced green fluorescent protein; VSV-G, vesicular stomatitis virus glycoprotein; DPP4, dipeptidyl-peptidase 4.

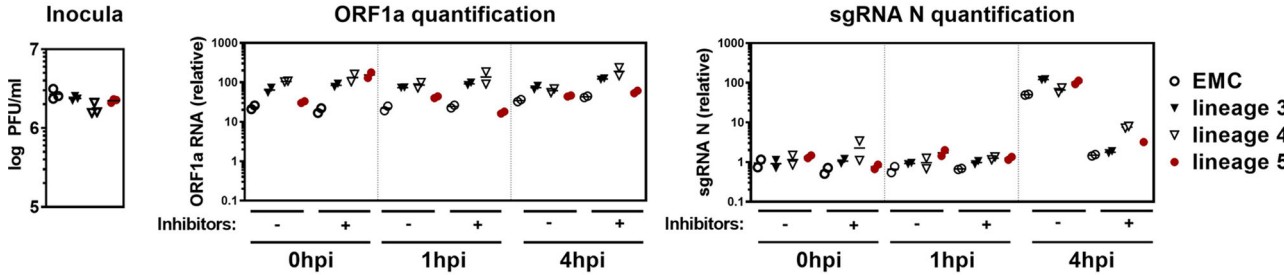

**Fig. 5 All tested MERS-CoV lineages show similar host cell entry.** Calu-3 cells were infected in duplicates with one virus isolate of each phylogenetic lineage (EMC, lineage 3, lineage 4, and lineage 5) at MOI = 1 in the presence or absence of CoV entry inhibitors. Entry inhibited samples were preincubated for 1 h with a cocktail of 25 µM Cathepsin L inhibitor, 25 µM Pitstop II, and 100 µM Camostat. The inhibitor cocktail remained on the cells during the whole course of infection. To allow for a synchronized virus entry, virus attachment was performed at 4 °C for 1 h, followed by four washing steps with PBS. After virus attachment, infected Calu-3 cells were either immediately lysed (0 hpi), or incubated at 37 °C for 1 or 4 h. Total RNA was isolated from lysed cells and viral genomic RNA (ORF1a) and subgenomic N mRNA (sgmRNA N) was quantified by q-RT-PCR and normalized to the housekeeping gene TBP. The inoculum of each phylogenetic lineage was back-titrated by plaque assay to confirm that highly similar amounts of infectious particles were used to infect the cells. PFU, plaque forming units; sgmRNA N, subgenomic nucleocapsid mRNA; ORF, open reading frame; hpi, hours post infection.

immune activation by viral antagonists[35]. Immune gene mRNA induction, in general, seemed to be highest with lineage 3 strains.

**IFN sensitivity.** Previous studies have shown that MERS-CoV EMC is highly sensitive towards type I IFN pretreatment[26]. To analyze IFN sensitivity among the different MERS-CoV lineages,

we compared virus replication in Calu-3 cells pretreated with IFN-I for 16 h. We monitored virus replication under low unit IFN concentrations that induce mainly an upregulation of cellular helicases (2.5 units IFN), as well as under IFN concentrations high enough to induce a full antiviral state by transcriptional upregulation of ISGs (25 units).

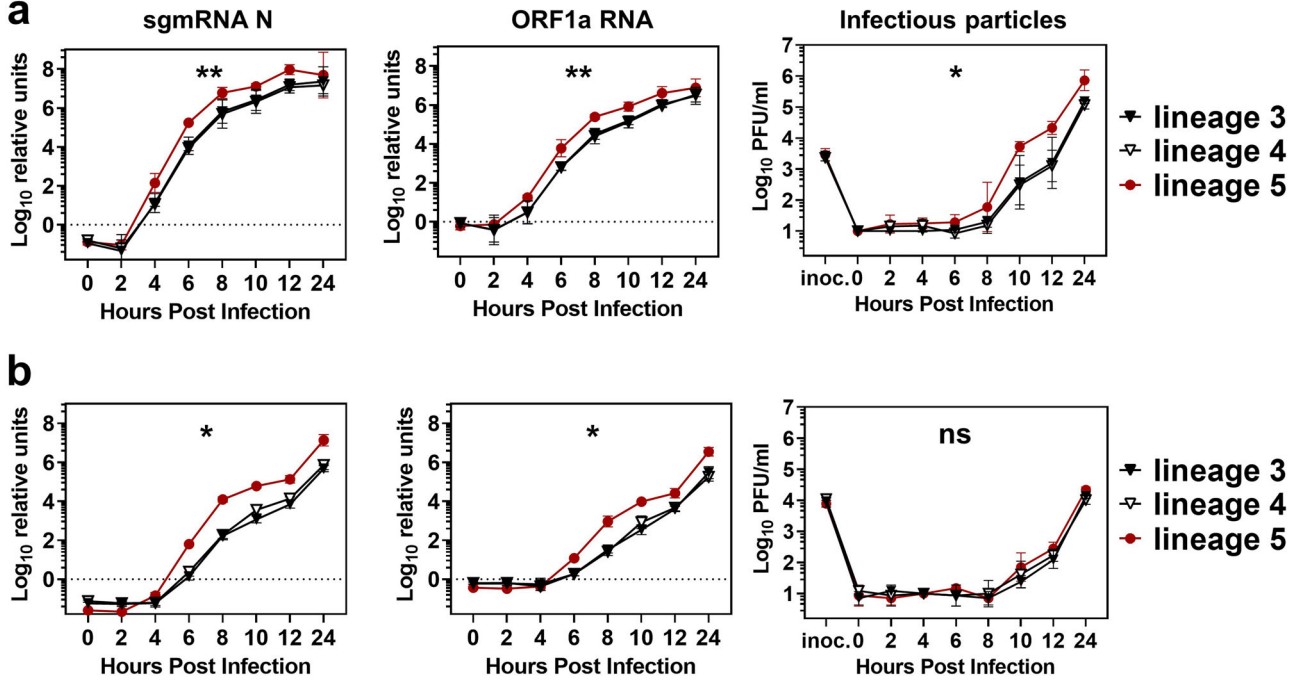

**Fig. 6 Subgenomic nucleocapsid RNA, ORF1a, and PFU quantification at early stages of infection. a** Calu-3 and **b** Vero B4 cells were infected at an MOI of 0.002 with one virus isolate of each phylogenetic lineage in biological duplicates for each indicated time point in $n = 2$ independent experiments. At the indicated time points, cell culture supernatant was harvested and subjected to virus quantification by plaque assay. Infected cells were lysed and total RNA was isolated and subjected to quantification by q-RT-PCR of sgmRNA N and ORF1a RNA, normalized to the housekeeping gene TBP, and the amount of RNA at 0 hpi. Shown are the combined data of two independent experiments. Statistical significance of variance in virus replication between groups was analyzed by one-way ANOVA. Significant differences were found for sgmRNA N (in **a**) Calu-3: **; $p = 0.0026$; in **b**) Vero B4: *; $p = 0.0276$), for ORF1a RNA (in **a**) Calu-3: **; $p = 0.0035$; in **b** Vero B4: *; 0.0418) and for PFU/ml in Calu-3 (*; $p = 0.0139$) PFU, plaque forming units; sgmRNA N, subgenomic nucleocapsid mRNA; ORF, open reading frame.

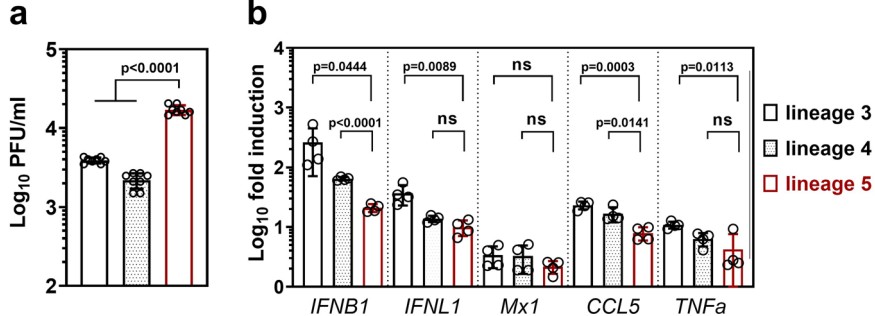

**Fig. 7 Lineage 5 MERS-CoV strains show reduced immune gene induction.** Calu-3 cells were infected with $n = 2$ isolates of each phylogenetic lineage with a high multiplicity of infection (MOI = 2) for subsequent quantitative real-time RT-PCR (q-RT-PCR) analysis of key immune genes. **a** Infectious virus production at 12 hpi was quantified by plaque assay. Shown are the results of two independent experiments with biological duplicates of $n = 2$ virus isolates, with bars indicating the mean and error bars indicating SD. **b** Immune gene induction as quantified by q-RT-PCR on lysed Calu-3 cells 12 hpi, expressed as fold induction over the noninfected control cells, normalized to the housekeeping gene TBP. The mean mRNA inductions of biological duplicates of $n = 2$ independent experiments with $n = 2$ virus isolates per lineage is shown with bars indicating the mean and error bars indicating SD. Statistical significance of differences in virus replication and fold inductions between lineage 5 and lineage 3 or 4 were analyzed by two-tailed, unpaired t-tests. PFU, plaque forming units; IFNB1, interferon beta 1; IFNL1, interferon lambda 1; CCL5, C-C motif chemokine ligand 5; TNFa, tumor necrosis factor-alpha.

As expected[36], replication of the reference strain EMC was already suppressed by low-level IFN pretreatment. Lineage 3 strains showed similarly high sensitivity, while 2.5 units of IFN did not detectably suppress lineage 4 and lineage 5 strains (Fig. 8). These results indicate that MERS-CoV lineages may differ in their ability to escape innate immune sensing, with a tendency for lineage 4 and 5 viruses to exhibit a more efficient escape under low unit IFN treatment. Twenty-five units of IFN markedly reduced virus replication of all lineages. The reduction

of replication caused by this dose was lower for lineage 5 than strains pertaining to lineage 3 and 4 (6.5-fold versus 23- and 18-fold, respectively).

**Influence of IFN action via the JAK/STAT signaling pathway.** To investigate if differences in virus replication are linked to viral suppression of IFN-signaling, we infected Calu-3 cells in the presence of the JAK/STAT inhibitor Ruxolitinib (Invivogen) (Fig. 9a), and compared virus replication in MERS-CoV

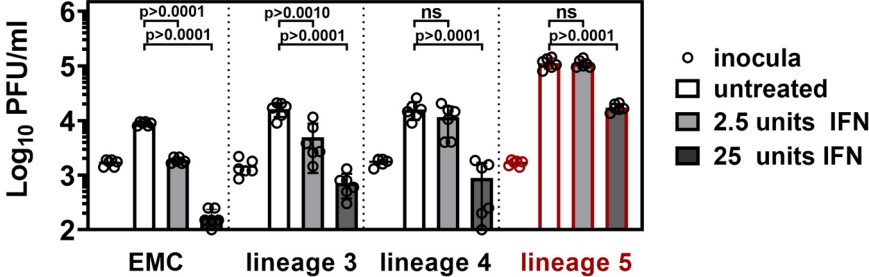

**Fig. 8 Lineage 5 MERS-CoV isolates show decreased IFN sensitivity.** Calu-3 cells were incubated for 16 h with 2.5 or 25 units type-I interferon (IFN) prior to infection at $MOI = 0.002$ with MERS-CoV isolates of the indicated lineages. Inoculum and virus progeny in culture supernatants were quantified by plaque assay 24 hpi. Infections were performed in biological triplicates with $n = 2$ virus isolates per lineage and the experiment was repeated three times. Shown is the mean PFU/ml of $n = 3$ independent experiments with bars indicating the mean and error bars indicating SD. Statistical significance of differences in virus replication between lineage 5 and lineage 3 or 4 was analyzed by two-tailed, unpaired $t$-tests. PFU, plaque forming units; IFN, interferon.

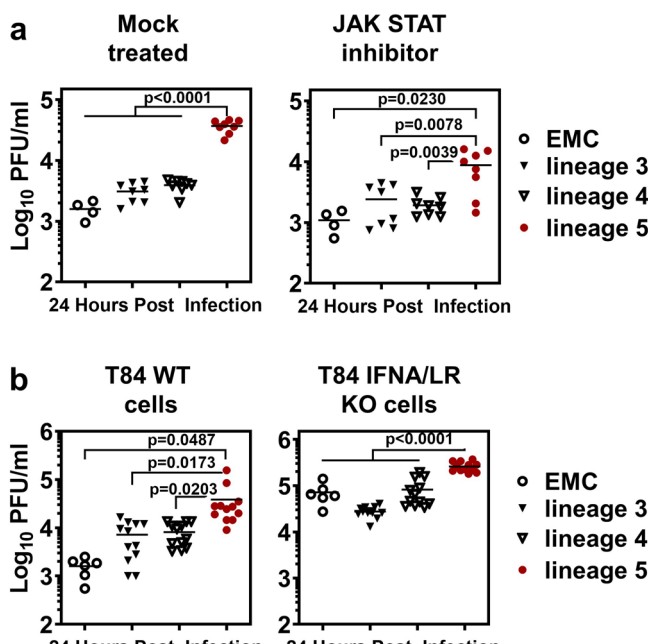

**Fig. 9 Replication of MERS-CoV isolates in JAK/STAT signaling inhibited and IFN receptor knockout cells.** Higher replication of lineage 5 MERS-CoV is additionally linked to IFN action via the JAK/STAT signaling pathway. **a** Calu-3 cells were treated with 50 nM of the JAK/STAT inhibitor Ruxolitinib 1 h prior and during the infection with viruses of the indicated phylogenetic lineages. Virus progeny in the culture supernatant was quantified by plaque assay at 24 hpi. Infections were performed in biological duplicates with $n = 2$ virus isolates per lineage and the experiment was repeated two times. Statistical significance of differences in virus replication between lineage 5 and lineage 3 or 4 or EMC was analyzed by two-tailed, unpaired $t$-tests. **b** T84 wt and type I and III IFN receptor knockout cells (IFNAR/IFNLR KO cells) were infected with viruses of the indicated phylogenetic lineages and virus progeny in the supernatant was quantified by plaque assay at 24 hpi. The infections were performed in biological duplicates with $n = 2$ virus isolates per phylogenetic lineage and the experiment was repeated two times. Statistical significance of differences in virus replication between lineage 5 and lineage 3 or 4 was analyzed by two-tailed, unpaired $t$-tests. PFU, plaque forming units; JAK, Janus kinase; STAT, signal transducer and activator of transcription; IFN, interferon; IFNA/LR KO, IFN-alpha- IFN-lambda-receptor double knockout; wt, wildtype.

susceptible cells carrying an IFN-alpha and -lambda receptor double knock-out (T84 cells, Fig. 9b). Whereas control experiments confirmed the higher replication levels of lineage 5, both approaches to IFN-signaling inactivation caused lineage 5 strains to only lose a part of their replicative prominence. Overall, lineage 5 strains retained a significantly higher replication level than strains pertaining to lineage 3 and 4, suggesting that differences in IFN-signaling antagonism alone do not explain the observed differences.

**Increased replication of lineage 5 MERS-CoV may not be determined by distinct mutations in previously described IFN antagonists PLP, p4a, p4b, and p5.** In order to explore which amino acid changes might be responsible for the observed increase of replication of lineage 5 isolates, we analyzed the amino acid divergence between virus isolates used in this study. We screened alignments for amino acid exchanges that are shared in isolates belonging to lineage 5 from 2015 and 2017 but absent in isolates of other phylogenetic lineages. In total, 16 amino acid exchanges were exclusively identified in lineage 5 isolates (Supplementary Table 3, highlighted in bold). Due to the phenotypic implication with IFN antagonism, we focused on amino acid exchanges in previously described IFN antagonists. We identified several lineage 5-specific mutations in nsp3 (A187V, A192V, A383T, E982A, A1150V, M1266I, and T1573I), but found no amino acid exchanges in the nsp3 papain-like protease (PLPro)[37]. There was a single amino acid exchange (A383T) in the nsp3 macrodomain that is thought to influence IFN antagonism in SARS- and related CoVs by an unknown mechanism[38,39].

Protein 4a and protein 5 are completely conserved between the three MERS-CoV phylogenetic lineages. We identified one lineage 5-specific amino acid exchange in protein 4b (p4b) at position 6 (M6T). Deletion of p4b has been shown to increase IFN induction in the context of virus infection[40]. We, therefore, employed our reverse genetics system[41] to generate a p4b M6T mutant virus. However, recombinant wild-type rEMC-CoV and rEMC-p4b-M6T viruses did not differ in replication (Supplementary Fig. 5).

**Replication phenotype in models of the human and camel respiratory tract.** Finally, to more closely reflect virus replication in the human respiratory tract, we infected differentiated human airway epithelia (HAEs) with two representative isolates from each phylogenetic lineage. According to our previous observations[26], we sampled supernatants exclusively from the apical site of differentiated HAE and quantified virus progeny every 24 h for four subsequent days. Lineage 5 isolates reached average titers up to 15-fold higher compared to lineage 3, lineage 4, and the EMC reference strain, with significant differences at 24 and 48 hpi (Fig. 10a). To provide a model of infection that most closely resembles infection of the human lung, we performed ex-vivo infections in human lung explants, derived from patients

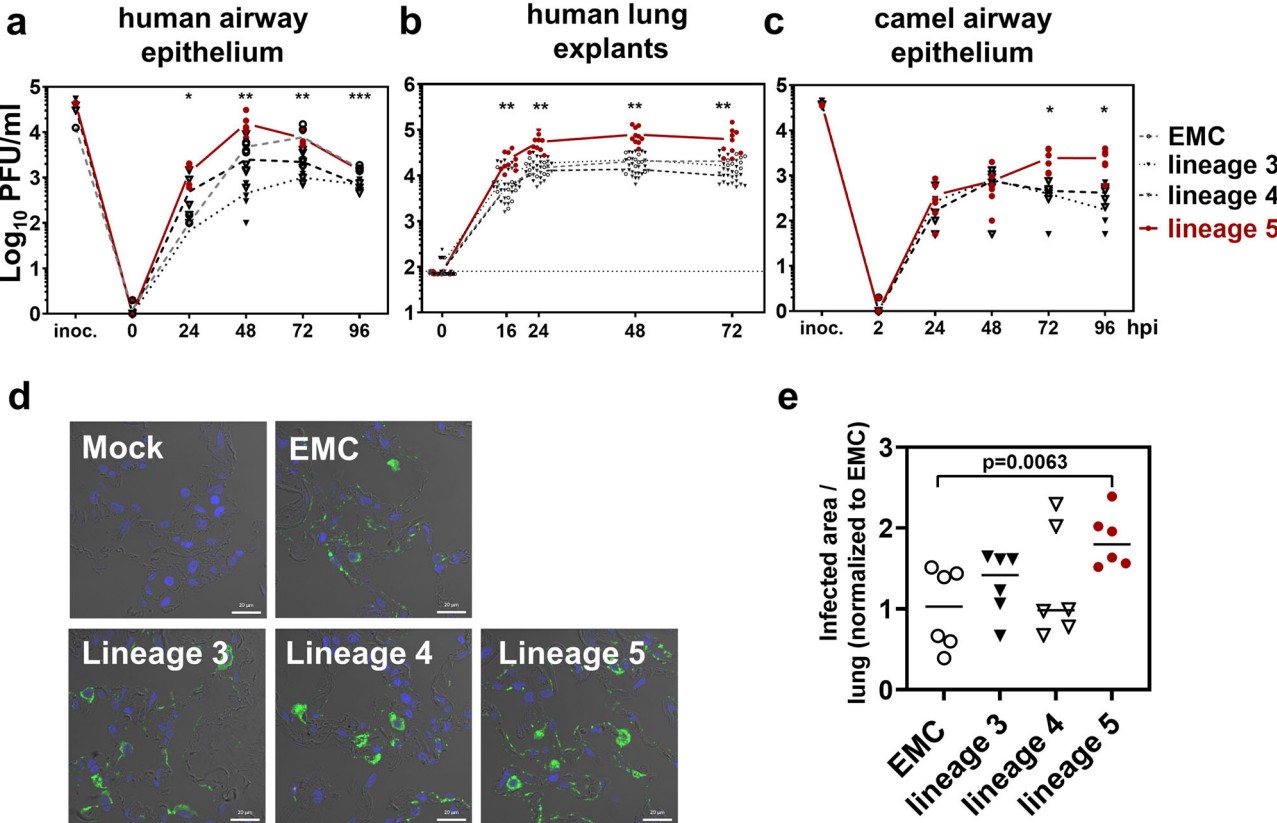

**Fig. 10 Growth kinetics of MERS-CoV isolates in models of the human and camel respiratory tract. a** Replication on primary human airway epithelium (HAE) of a single donor. Two isolates of each phylogenetic lineage were used for infection in biological triplicates. Virus progeny in apical washes was quantified every 24 h by plaque assay. Statistical significance in difference of PFU/ml between lineage 5 and other lineages was determined by one-way ANOVA and two-tailed Student's $t$-test for each time point individually. Statistically significant differences in PFU/ml determined by one-way ANOVA corrected for multiple comparisons using the Tukey method were found at 24 hpi (lineage 5 vs. lineage 3: *; $p = 0.0274$; lineage 5 vs. EMC: *; $p = 0.0301$); at 48 hpi (lineage 5 vs. lineage 3: *; $p = 0.0248$; lineage 5 vs. lineage 4: *; $p = 0.0398$); at 72 hpi (lineage 5 vs. lineage 3: **; $p = 0.0095$; lineage 5 vs. lineage 4: *; $p = 0.0190$) and at 96 hpi (lineage 5 vs. lineage 3: ****; $p < 0.0001$; lineage 5 vs. lineage 4: *; $p = 0.0123$; lineage 5 vs. EMC: **; $p = 0.0050$) **b** Replication of MERS-CoV isolates on ex vivo lung explants, derived from $n = 3$ different patients that have undergone lung resection. One isolate of each phylogenetic lineage was used for infection in biological triplicates for each explant and virus progeny in the supernatant was quantified by plaque assay. Differences in PFU/ml between lineage 5 and other lineages was tested for significance using the two-sided Kruskal–Wallis test. Statistically significant differences in PFU/ml were found at 16 hpi (lineage 5 vs. EMC: **; $p = 0.0024$; lineage 5 vs. lineage 4: **; $p = 0.0012$), at 24 hpi (lineage 5 vs. EMC: ***; $p = 0.0006$; lineage 5 vs. lineage 3: **; $p = 0.0080$; lineage 5 vs. lineage 4: ****; $p < 0.0001$), at 48 hpi (lineage 5 vs. EMC: **; $p = 0.0064$; lineage 5 vs. lineage 3: **; $p = 0.0034$; lineage 5 vs. lineage 4: ****; $p < 0.0001$) and at 72 hpi (lineage 5 vs. lineage 3: **; $p = 0.0029$; lineage 5 vs. lineage 4: ****; $p < 0.0001$) **c** Replication of MERS-CoV isolates on well-differentiated primary Bactrian camel airway epithelium. Two isolates of each phylogenetic lineage were used for infection in biological triplicates. Virus progeny in apical washes was quantified every 24 h by plaque assay. Statistical significance in difference of PFU/ml between lineage 5 and other lineages was determined by one-way ANOVA and two-tailed Student's $t$-test for each time point individually. Statistically significant differences in PFU/ml were found at 72 hpi (lineage 5 vs. lineage 3: *; $p = 0.0222$; lineage 5 vs. lineage 4: *; $p = 0.0266$ and at 96 hpi (lineage 5 vs. lineage 3: *; $p = 0.0144$; lineage 5 vs. lineage 4:*; $p = 0.0244$). **d** Immunofluorescent analysis of MERS-CoV lineages replication in ex vivo human lung tissue. Explants from $n = 3$ different patients were infected with one MERS-CoV isolate of each lineage for 24 h. Histological sections were probed with MERS-CoV nucleocapsid antibody (green), nuclei were counterstained with DAPI (blue), and tissue structure was visualized by differential interference contrast. Scale bar represents 20 μM. **e** MERS-CoV replication quantified by immunofluorescent analysis. Values represent the area (in μm²) of positive-infection, determined by nucleocapsid immunofluorescent signal in $n = 3$ infected lung explants at 24 hpi, measured in biological duplicates. The complete piece of each lung explant was analyzed for quantification. The positive-infected area of each respective isolate was normalized to the total lung tissue area and the value obtained for the EMC strain-infected lung tissue. Statistical significance in a quantified infected area between lineage 5 and EMC was analyzed by two-tailed, unpaired $t$-tests. Scale bar = 20 μm. PFU, plaque forming units.

that have undergone lung resection. We used one representative virus isolate per phylogenetic lineage to infect lung explants derived from three different donors (Fig. 10b). Supernatants of infected lung explants were harvested at 16, 24, 48, and 72 hpi. Similar to HAEs, lineage 5 titers were up to fivefold higher compared to titers of lineage 3, lineage 4, and the EMC reference strain, with significant differences at 24 and 48 hpi (Fig. 10b). Higher ex-vivo replication of lineage 5 viruses was confirmed by

immunofluorescent analysis of infected lung explants (Fig. 10d). We quantified the area staining positive for MERS-CoV infection in three lung explants, derived from three donors, and found averages of infected areas to be highest in samples infected with MERS-CoV lineage 5, at 24 hpi (Fig. 10e). We next infected well-differentiated Bactrian camel airway epithelium cells with two representative isolates of each phylogenetic lineage. We found lineage 5 MERS-CoV isolates to show significantly increased virus

titers at 48 and 72 hpi (Fig. 10c). This indicates that increased replication of lineage 5 seems to be independent of whether the virus replicates in human or camel tissue.

## Discussion

The present study demonstrates that a newly emerged genetic lineage of MERS-CoV has higher replicative fitness and causes a lower level of IFN and cytokine induction than previously circulating strains. An example of recombination between coronavirus lineages that gave rise to a more pathogenic strain has already been provided by the cases of feline coronaviruses and avian infectious bronchitis virus[42,43]. To our knowledge, the present study is the first to demonstrate a phenotypic change through recombination in a CoV infecting humans. Its occurrence in MERS-CoV is particularly relevant from a public health perspective as the current human-to-human transmission rate during sporadic MERS-CoV outbreaks is close to the critical threshold for sustained transmission ($R_0 = 0.6–1$)[6,7]. Since 2015, lineage 5 was highly prevalent in sampled dromedary camels and humans in Saudi Arabia[19,20], and caused a major outbreak in South Korea[14–17]. This upsurge triggered speculations as to the transmissibility or fitness of the novel virus variant[19,20]. Nevertheless, the phenotype of these viruses, as opposed to viruses circulating earlier, has never been studied. The present study provides a first comprehensive phenotypic assessment of lineage 5 isolates, showing increases in replicative fitness over previously circulating lineages.

The in vitro and ex vivo cell culture models used in our study may not necessarily reflect virus excretion in vivo, but the observed increase in fitness in these models agrees with epidemiological trends over several years, in a large geographic region. Even slight increases in replication may have a profound long-term influence on pathogen fitness in the host population. As the increase in replication was also observed in human-derived systems, increased virus shedding by lineage 5 in humans is conceivable. Because the transmitted virus dose is positively correlated with the rate of adaptive evolution, lineage 5 may pose an increased risk of human adaptation when transmission chains occur in nosocomial outbreaks[44,45].

The here reported functional diversity of circulating MERS-CoV on the Arabian Peninsula is somewhat surprising given that earlier studies found little indication for phenotypic differences between viral lineages[17,24,46]. A study of African MERS-CoV isolates found functional diversity[12] that could be attributed to deletions in ORF4b encoding a suppressor of RNAse L that acts via an active phosphodiesterase function[27,40]. However, all MERS-CoV isolates used in this study have a full gene repertoire thus providing little angle to link changes of phenotype to single viral proteins. Our data indicate that the increased fitness of MERS-CoV lineage 5 does not result from changes affecting viral entry and is unlikely to be based on an altered interference with IFN-signaling. Differences in IFN sensitivity and ISG induction may be secondarily caused by the primary differences in cytokine induction.

Multiple viral proteins have been associated with the modulation of IFN and cytokine induction in MERS- and related CoVs infected cells, including nsp1, nsp3, nsp14, nsp15, nsp10/16, p4a, and p4b[29,30,38,40,47–52]. Among the present MERS-CoV lineages most of these proteins are either fully conserved (nsp16, ORF4a, and ORF5), conserved in functional domains (nsp3 PLPro and macrodomain, nsp14 N7-MTase domain, and nsp15 EndoU domain) or show amino acid substitutions unlikely to affect protein function mediated by changes in polarity or charge, e.g., F158V in nsp1 (Supplementary Table 4)[50,51,53,54].

We particularly focused on amino acid positions in viral proteins for which structural data are available (PLPro in nsp3) and on proteins uniquely attributed to IFN antagonism in MERS-CoV (p4a and p4b). The PLPro in nsp3 of most CoVs encodes for a deubiquitinating domain, which seems to counteract the antiviral function of ISG15, and importantly, the transcriptional activity of IRF3, antagonizing IRF3-mediated induction of IFNs, and cytokines downstream of virus sensing[48,49,55–57]. The second motif in nsp3, the macrodomain, may further contribute to nsp3-mediated immune antagonism in SARS- and related CoVs by its ADP-ribose-1′-phosphatase domain that seems to decrease IFN and ISG induction in infected cells by a yet unknown mechanism[38]. However, all divergent amino acids lay outside of the macrodomain and described functional residues of PLPro[37,58]. MERS-CoV accessory protein 4a antagonizes IFN induction by binding to dsRNA and inhibiting MDA5 activation, possibly by binding to its activator PACT[29,59]. However, no lineage-specific amino acid exchanges are present in p4a of lineage 5 MERS-CoV. There was one potentially relevant exchange in p4b, a functional antagonist of RNase L. We tested the effect of this exchange by introducing it into a MERS-CoV reference strain via reverse genetics[41]. However, the change did not influence the replication level.

It is conceivable that additive mutations combined from donor lineages by recombination may cause the fitness increase. Fitness-increasing secondary mutations could also have occurred after the recombination event. There is some evidence from our data that members of the parental lineage 4 show an attenuated cytokine-based immune response, but these viruses do not show increased fitness overall. Complementing mutations in the recombinant virus could be mapped by shuffling of gene portions through reverse genetics. This may help to understand whether and how cooperating mutations may have been assembled into one genome following recombination, and how the recombinant virus phenotype was selected for. Mapping those causal mutations to functional domains within the viral genome might help to understand which viral proteins contribute to the pathogenicity of newly emerging MERS-CoV.

Based on the present study, we can only resolve the mechanism of fitness increase to the level of different compartments of cellular immunity, identifying cytokine and IFN induction as a discriminative trait. It is a limitation that deeper mechanistic investigation cannot be provided. We note that the observed differences in virus–host interaction may originate outside the direct infection sensing and IFN induction mechanisms so that changes in IFN and cytokine induction, as well as the partial differences in IFN sensitivity, might be collateral effects.

Our study is limited in that we cannot provide experimental data on the replication phenotype of MERS-CoV isolates circulating later than 2017. Given that lineage 5 MERS-CoV remained the dominant lineage in dromedary camels through 2018[19], it seems unlikely that changes to its relevance in humans have occurred. Recent declines in case notification and full genome sequence deposition might be caused by the COVID-19 pandemic that has absorbed a lot of surveillance capacities. Moreover, it should be noted that the observed increased replication of lineage 5 MERS-CoV does not implicate virus selection in humans. Analyses of viral populations have led to the conclusion that human-to-human transmission does not play a relevant role in MERS-CoV evolutionary dynamics, and selection therefore will have taken place in the animal reservoir[60]. However, the reduced induction of cytokine expression is compatible with the selection of lineage 5 in dromedary camels as immune sensors and appending signal transduction cascades triggering cytokine induction are conserved among mammals. Our experiments

using differentiated camelid airway cultures indeed confirm a replicative advantage for lineage 5 viruses that is similar to replication in human cells. Selection for virulence or immune escape in dromedary camels may thus involve a collateral benefit for the virus once transmitted to humans. In light of an ongoing SARS-CoV-2 pandemic, the emergence and epidemiological dominance of a more replicative strain of MERS-CoV warrants an increased surveillance of circulating MERS-CoV strains.

## Methods

**Viruses**. All virus stocks have been primarily isolated from respiratory samples, obtained from Saudi Arabian patients diagnosed with MERS between March 2014 and November 2015. All patient samples were obtained for the purpose of outbreak investigations by the KSA Ministry of Health and were thus exempt from institutional review board oversight. A description of the study conditions is provided in ref. [25]. Samples for the present study were utilized without knowledge or usage of person-related data. Samples with a q-RT-PCR CT value higher than 25 were selected for primary virus isolation as described in ref. [25]. Successful primary isolates were used for high titer stock production in Vero B4 cells and were concentrated using Vivaspin columns (100,000 MWCO PES membrane; Sartorius) upon reaching maximal cytopathic effect after daily monitoring. All stocks were quantified using plaque titration as described below.

**Virus sequence analysis**. Viral RNA of 22 passage two virus stocks was extracted using the Machinery and Nagel viral RNA kit. Viral RNA was subjected to cDNA synthesis using the SuperScript® One-Cycle cDNA Kit (Invitrogen) according to the manufacturer's protocol. cDNA was fragmented and supplemented with adapter sequences using the Nextera® XT DNA Library Prep (Illumina), subsequently processed with the MiSeq Reagent Kit v3 (Illumina) and subjected to NGS on a MiSeq™ System (Illumina), all according to the manufacturer's protocol. Using Geneious software, version 9.1.8, contigs were assembled by mapping reads to the EMC genome (Genbank ref.: JX869059). Assembled full genomes have been deposited to NCBI GenBank under the accession numbers MN481964 to MN481978.

**Phylogeny and recombination analysis**. Phylogenies of whole-genome and subgenomic sequence alignments were estimated by the maximum-likelihood approach, implemented in IQtree, version 2.0.3 (whole genome alignment)[61], or RaxML, version 8.0.0 (subgenomic genome alignments). Whole-genome phylogenies were estimated with a transition model and unequal base frequency (TIM + F + I) and 1,000 bootstrap replicates. RaxML trees were estimated with a general time-reversible (GTR) model of nucleotide substitution with estimated gamma-distributed rate variation and a proportion of invariant sites. About 10,000 bootstrap replicates were run. Input alignments of whole-genome and subgenomic sequences comprising the outer and inner nonrecombinant segments were aligned by implementing ClustalW in Geneious software, version 9.1.8.

For detailed recombination analysis, we performed a BootScan analysis with 500 bootstraps in SimPlot, version 3.5.1., with a sliding window of 5,000 and 200 bp steps.

**General cell culture procedures**. All cells were maintained in Dulbecco's Modified Eagles Medium (DMEM) at 37 °C, 5% CO$_2$ as described earlier[25]. For virus titration, Vero B4 cells were used. For infection studies Vero B4 (DSMZ-AC33), Calu-3 (ATCC HTB-55) and Caco-2 (ATCC HTB-37), T84 (ATCC-CCL-248), and T84 IFN alpha and lambda receptor double knockout cells were used. For polarization, Calu-3 cells were seeded on transwell inserts (Greiner Bio, pore size 0.4 μm) and initially cultivated under liquid-liquid conditions for 3 days. Subsequently, cells were transferred to the ALI condition with basolateral medium exchange every 2 days for 14 days. Prior to infection, complete polarization was confirmed by transepithelial electrical resistance measurement. For VSVpp transduction studies, 293 T, Caco-2, Calu-3, and Huh-7 cells were used and maintained as described above. Well-differentiated human and Bactrian camel airway epithelial cell cultures were generated and maintained as described in ref. [62]. In brief, primary human tracheobronchial epithelial cells were isolated from patients (>18 years old) undergoing bronchoscopy or pulmonary resection at the Cantonal Hospital in St. Gallen, Switzerland, in accordance with ethical approval (EKSG 11/044, EKSG 11/103, KEK-BE 302/2015). Bactrian camel tracheobronchial epithelial cells were isolated from postmortem tissue obtained from the veterinary hospital of the University of Bern. To generate well-differentiated AEC cultures, cryopreserved cells were thawed and expanded into collagen type I-coated flasks in Bronchial Epithelial Growth Media (BEGM,) medium, followed by seeding into collagen type IV-coated porous inserts (6.5 mm radius insert, Costar, Corning) in 24-well plates. After seeding, cells were grown at a liquid-liquid interface for 2–3 days in a BEGM medium. Once cells reached confluency, cultures were air-lifted to establish an ALI to allow for cellular differentiation using the ALI medium. For Bactrian camel AEC cultures, the EGF concentration in the ALI medium was increased to 5 ng/ml. Both human and Bactrian camel AEC cultures were maintained at 37 °C in a humidified

incubator with 5% CO2 and the basolateral ALI medium was changed every 2–3 days.

**Lung explants infection, culture procedure, and immunofluorescent analysis**. Lung explants were obtained from patients undergoing lung resection. Written informed consent was obtained from all patients and the study was approved by the ethics committee at Charité clinic (project EA2/079/13). For each experiment tumor-free human lung tissue was cut into small pieces (weight approx. 0.1–0.2 mg per piece) and incubated in RPMI 1640 medium at 37 °C with 5% CO2. After overnight incubation lung organ cultures were inoculated with 1x10e5 PFU of virus for 1.5 h under shaking conditions followed by one washing step with phosphate-buffered saline (PBS) to remove the excess virus. Lung tissue was incubated for up to 72 hpi in RPMI 1640 medium containing 10% fetal calf serum and 2 mM L-glutamine. For growth curve analysis supernatants of infected lung tissue were harvested at 0, 16, 24, 48, and 72 hpi and supernatants were titrated on Vero E6 cells by standard plaque titration assay.

For immunofluorescent analysis, processing of human lung tissues was performed as described in ref. [63]. In brief, MERS-CoV infected tissue samples were fixed in 3% paraformaldehyde twice for 24 h. Samples were subjected to anti-MERS-CoV nucleocapsid primary antibody (Sino Biological, China) detection overnight at 4 °C and labeled with a secondary antibody Alexa Fluor 488 (Invitrogen). Nucleus counterstaining was performed with DAPI (Sigma Aldrich). Immunofluorescence spectral confocal microscopy was conducted using an LSM 780 [(objectives: Plan Apochromat 63x/1.40 oil DIC M27), Carl-Zeiss]. Based on a spectral image lambda stack, linear deconvolution of tissue autofluorescence and overlapping spectra of fluorochromes was performed using ZEN 2012 software (Carl-Zeiss). Each image shows a maximum fluorescence intensity projection of a z-stack representing the entire 10 μm section. To reveal lung and cell morphology, images were combined with Differential Interference Contrast (DIC). All image sets were acquired using equal configurations. Images were processed using ZEN 2012 (Carl-Zeiss)and ImageJ software (http://imagej.nih.gov/ij/). Quantification of Alexa Fluor 488 signal intensity was performed using ZEN 3.1 (blue edition; Carl-Zeiss). Briefly, stained human lung slides were scanned using Axio scan.Z1 (Carl-Zeiss) followed by digital image segmentation of positive-infected areas in μm$^2$ and normalization to the total area of the lung slices, measured by autofluorescence of the lung tissue.

**Virus infections**. Vero B4 and Caco-2 cells were seeded at 1.5x10e5 cells/well, Calu-3 cells at 3x10e5 cells/well, and T84 cells at 2x10e5 cells/well, all in composite DMEM in 24-well plates 16 h prior to infection. Cultured cells (Vero B4, Calu-3, Caco-2, and T84 cells) were infected with passage two MERS-CoV stocks diluted to the desired MOI in OptiPro at 37 °C. Primary airway epithelia (AEs) were infected with 4,000 PFU in Hanks Balanced Salt Solution (HBSS) for 1.5 h. Subsequently, cells were washed three times with PBS or HBSS for AEs, respectively, and further incubated. At 24 and 48 h, respectively 24, 48, 72, and 96 h for AEs, supernatants were harvested and viral loads were quantified by plaque titration on Vero B4 cells as described below. For IFN priming experiments, Calu-3 and Vero B4 cells were treated 24 h post seeding with either 0, 2.5, or 25 units pan-species type I IFN (Biochrom) for 16 h prior to infection.

**Synchronized infections and virus–host cell entry blocking**. For virus cell entry experiments, Calu-3 cells were seeded 16 h before infection in a 24-well format at a seeding density of 3x10e5 cells/well. One hour prior to virus infection, cells were washed twice with PBS and treated with composite DMEM without FCS, containing only DMSO (no inhibitors) or DMEM without FCS, containing a triple combination of previously described CoV entry inhibitors resolved in DMSO (inhibitor samples). Cathepsin L inhibitor MDL28170 and Pitstop II (both Sigma-Aldrich) were used at a final concentration of 25 μM, Camostat mesylate (Sigma-Aldrich) was used at a final concentration of 100 μM. Calu-3 cells infection was performed on ice and the cells were immediately put on 4 °C for 1 h after pipetting virus dilutions onto the cells to allow for virus attachment for the cells without receptor-mediated virus entry. Cells were subsequently washed five times with PBS to remove the majority of unbound virus particles and either immediately lysed (0 hpi samples), or incubated with composite DMEM, with or without the inhibitor cocktail, for 1 or 4 more hours post infection (1 hpi and 4 hpi samples), prior to medium removal and cell lysis. RNA isolation of cell lysates and quantitative real-time PCR was performed as mentioned below.

**Plaque titration and plaque reduction neutralization test**. Vero B4 cells were seeded at 1.5x10$^5$ cells/well in 24-well plates 16 h prior to titration. A 1:2 mixture of Avicel (2.4 g/l, FMC BioPolymer) and double concentrated DMEM was used as an overlay. Four days post infection, cells were washed, fixed, and stained in 6% formaldehyde solution containing crystal violet. Plaque reduction neutralization tests (PRNT) was performed as described in ref. [25]. In brief, 25 PFU of each isolate were incubated with serially diluted serum for 1 h at 37 °C, prior to plaque titration. The following five sera were used for neutralization: Munich-1 (MERS patient, Germany 2014), SA278 (MERS patient, KSA 2014), Dubai-S1 (dromedary camel, UAE 2014), Kenia-ILRI (dromedary camel, Kenya 2017), and Pakistan-493 (dromedary camel, Pakistan 2017) (Supplementary Table 2).

*Cloning of lineage-specific Spike genes and generation of VSV pseudotype particles (VSVpp) and transduction of target cells.* Previously cloned pCG1 vector carrying the EMC spike gene was subjected to mutagenesis PCR to accommodate the three amino acid substitutions specific for lineage 3, lineage 4, and lineage 5 (Supplementary Fig. 3). In the first round of mutagenesis PCR using Phusion polymerase (Thermo Fisher Scientific), the Q1020R mutation was introduced (all lineages). The obtained plasmid was then subjected to the second round of mutagenesis PCR, introducing either L411F (lineage 3) or Q833R (lineage 5). 5′phosporylated primers were used for mutagenesis PCR. indicated in Supplementary Table 5 with the underlined nucleotides being the substitutional sequences.

Obtained pCG1 plasmids were used for the production of rhabdoviral spike proteins pseudotyped vectors, generated according to a previously published protocol[21]. Briefly, 293 T cells transfected with pCG1 vectors, carrying EMC-Spike, MERS-S Q1020R, MERS-S Q1020R/L411F, MERS-S Q1020R/Q833R, VSV-G (positive control), or empty expression plasmid (negative control) were inoculated with replication-deficient, VSV-G-*trans* complemented VSV in which the VSV-G gene was replaced by eGFP and firefly luciferase, VSV*ΔG (VSV-G)[64] that was kindly provided by Gert Zimmer. After 1 h incubation at 37 °C, the supernatant was removed, the cells were washed with PBS and a fresh culture medium was added. In case of spike protein-expressing cells or cells transfected with an empty expression vector, the medium was supplemented with the VSV-G-neutralizing antibody I1 (produced from CRL-2700 cell line, ATCC).

For transduction of target cells in 96-well format, the culture medium was removed and cells were inoculated with 100 µl VSVpp. At 18 h post inoculation cells were lysed for 30 min at RT in 50 µl Luciferase Cell Culture Lysis Reagent (Promega) were added. Lysates were transferred into white, opaque-walled 96-well plates, and activity of virus-encoded luciferase was measured using the Luciferase Assay System substrate (Promega) and the Hidex Sense plate reader (Hidex).

**Quantification of MERS-S binding to DPP4 by flow cytometry.** Analysis of S protein binding to DPP4 was analyzed by flow cytometry employing a previously published protocol[21]. In brief, 293 T cells were untransfected or transfected with the overexpression vector pCG1 either empty or carrying MERS-CoV Spike protein of clade A (EMC), EMC I529T, lineage 3 (L411F/Q1020R), lineage 4 (Q833R/Q1020R), or lineage 5 (Q1020R), as well as an empty vector for negative control. At 48 h post transfection, cells were resuspended in PBS, centrifuged (5 min at 600xg at 4 °C), and resuspended in PBS containing 1% bovine serum albumin (1% BSA/PBS) for washing. The cells were centrifuged again and incubated in 1% BSA/PBS containing soluble DPP4 equipped with a C-terminal human Fc-tag (solDPP4-Fc,1:50, 1:200 and 1:1,000; ACROBiosystems) at 4 °C in an overhead shaker for 1 h. Afterward, the cells were pelleted and incubated in 1% BSA/PBS containing an Alexa Fluor488-conjugated antihuman antibody (1:500; Thermo Fisher Scientific), for 1 h at 4 °C in an overhead shaker. Subsequently the cells were pelleted, washed with 1% BSA/PBS, and fixed in 4 % paraformaldehyde solution for 20 min at room temperature. Prior to analysis via flow cytometry, the cells were pelleted again and washed with 1% BSA/PBS. Flow cytometry was conducted on an LSR II Flow Cytometer, with a double gating strategy (1) Total cells: SSC-A vs. FSC-A and (2) Singlets: FSC-H vs. FSC-A. Quantification of an Alexa Fluor488 signal was performed in the second gate. Data were further processed in the FCS Express 4 Flow research software (De Novo software).

**Direct competition assay.** Virus stocks of one lineage 5 isolate (Riyadh-1764-2015) and one lineage 3 isolate (Riyadh-1147-2014) were mixed at two ratios (1:1 and 1:9 lineage 5:lineage 3) and used to infect Calu-3 cells in duplicates at an MOI 0.04 (corresponding to 10,000 total PFU). As a control, a single infection with only 10,000 PFU of the lineage 5 or the lineage 3 isolate was performed. At 24 hpi, 100 µl of the cell culture supernatants of infected cells were used to inoculate new Calu-3 cells for five subsequent passages. Viral RNA was isolated from each inoculum (passage 0/p0) and after five passages (passage 5/p5) using the Machinery and Nagel vRNA kit and three short PCR amplicons were reverse transcribed (SSIII/Taq one-step kit; Thermo Fisher Scientific) using specific primers (Supplementary Table 5)The obtained PCR amplicons was sent for Sanger sequencing. For peak height analysis, the web-based Chromat Quanitator (Mullins lab, University of Washington) was used and quantified sanger sequencing peak heights were averaged for all duplicates and calculated as a percentile of the total height.

Generation of recombinant rMERS-CoV rEMC-p4b-M6T by red-mediated recombination A MERS-CoV strain carrying the lineage 5 specific ORF4b M6T mutation (rEMC-p4b-M6T) was constructed from our cDNA clone by red-mediated recombination[41]. We generated the I-*SceI*-*aphAI* transfer cassette containing 5′and 3′ 40 bp homologous hooks to the MERS-CoV EMC genome and the M6T mutation by Phusion PCR amplification (Thermo Fisher Scientific). Electroporation of the purified transfer construct and both recombination steps were performed as described in Muth et al., 2017[41]. Bacterial clones carrying the correct insert size after first and second recombination were identified by PCR. Virus rescue from purified full-length SARS-CoV cDNA clones was performed as described in Muth et al. 2017[41]. In brief, cDNA plasmids were linearized by MluI digestion, in vitro transcribed, and capped (mMESSAGE mMACHINE T7 Transcription Kit, Thermo Fisher Scientific). In vitro transcripts were electroporated into BHK-21 cells. 24 and 48 h post electroporation, 1 ml of the supernatant was transferred to susceptible Vero B4 cells and virus replication was

monitored by quantitative real-time PCR[65]. Recombinant viruses were harvested three days post infection. Virus stocks were purified and subjected to full-genome sequencing on a MiSeq System (Illumina).

**Quantitative real-time PCR.** For quantitative real-time PCR analysis, total RNA of MERS-CoV infected Calu-3 cells was harvested at 12 hpi using the Machinery and Nagel total RNA kit, following the manufacturer's instructions. A two-step PCR protocol was used for the quantification of host transcripts. First, cDNA was prepared from equal molar viral RNA using the iScript™ cDNA synthesis kit (Bio-Rad), according to the manufacturer's protocol. Second, cDNA was amplified and quantified using Taq-polymerase (Invitrogen) and a set of real-time primer and probes for each transcript under investigation for sgmRNA N and ORF1a RNA analysis, cells were harvested at 0, 2, 4, 6, 8, 10, 12, and 24 hpi and total RNA was isolated and subjected to quantitative real-time PCR using the one-step SuperScript III kit (Invitrogen) with primers published in ref. [65] and ref. [66].

**Statistical analysis.** Analysis of variance (ANOVA) with Bonferroni's post hoc test was used for group-wise comparisons. Unpaired Student´s *t*-test test was applied for comparisons of groups. Statistical tests were implemented in GraphPad Prism, version 8.2.1. All statistical tests were two-tailed and $p < 0.05$ was considered statistically significant, with asterisks indicating the degree of significance as follows: ns $P > 0.05$; *$P \le 0.05$; **$P \le 0.01$; ***$P \le 0.001$; and ****$P \le 0.0001$.

**Reporting Summary.** Further information on research design is available in the Nature Research Reporting Summary linked to this article.

## Data availability

All MERS-CoV genome sequences used in this study are available on GenBank under accession numbers MN481964 to MN481978 (Supplementary Table 4). Source data displayed in Figs. 2–10 and Supplementary Figs. 2, 3, and 5 are provided online (source data file).

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

## Acknowledgements

The work was supported by the Bundesministerium für Bildung und Forschung under Grant 01KI1723A (RAPID), as well as the European Union via Project ReCoVer (grant number GA101003589). M.A.M. received funding from the DFG Infectiology program (MU3564/3-1). M.A.M. and C.D. were supported by the United States National Science Foundation, award number 1816064. S.B. was supported by the Heisenberg program (project number 415089553) and M.L.S. was supported by the DFG (project number 41607209). M.G., R.D., and V.T. were supported by the European Commission (Marie Sklodowska-Curie Innovative Training Network HONOURS; grant agreement no. 721367). A.H. and S.H. were supported by DFG (SFB-TR 84) and Charite 3 R, additionally A.H by Charité-Zeiss MultiDim. We thank Abdulla M. Assiri, Public Health Directorate, Ministry of Health, Riyadh, for the provision of additional clinical samples. We thank PD Dr. Ulrich Wernery, Central Veterinary Research Laboratory, Dubai, UAE,

for providing essential material for this study and Jackson Emanuel (Charité) for editing the manuscript.

## Author contributions

S.S., M.A.M., and C.D. conceived and designed the experiments. S.S., C.M., H.K.-W., D.F., and A.R. performed experiments. S.S., D.M., and V.M.C. performed virus isolations from clinical samples. Z.A.M. provided clinical samples. M.L.S., S.B., R.D., M.G., S.E., A.H., S.H., and V.T. provided essential materials. S.S., C.D., and M.A.M. wrote the manuscript. S.S. prepared all figures.

## Funding

## Competing interests

The authors declare no competing interests.
