## [Peer Review File · Nature Communications]

Reviewer comments, first round –

Reviewer #1 (Remarks to the Author):

A comparison of various isolates belonging to different lineages of MERS-CoV is performed in a number of cell lines and identifies that virus isolates from lineage 5 appear to have a replicative advantage over EMC2012 and other lineage viruses. Interestingly, it appears that lineage 5, which also happens to be a recombinant virus, has taken over to become the predominant lineage in Saudi Arabia. Based on limited data provided it appears that it rapidly displaced all other lineages and now is the only lineage circulating. However, this finding is underpowered by a lack of samples from years 2016 onwards - somewhat limiting this conclusion, but that is not a major flaw. Attempts are made to identify a mechanism behind this fitness gain through competition assays, entry assays (various), binding assays, and various replication (genome and virus) assays (including in primary cells). None of these mechanisms appear to be the cause of this finding. The authors do however identify that lineage 5 viruses induce a lesser innate immune response despite increased (or because of) increased replication and that lineage 5 viruses are less sensitive to IFN-induced virus replication restriction.

The increased fitness effect is not that prevalent in most of the studies however it is consistent and possibly the strongest evidence is that virus competition assays where even at a ratio of 9:1, the lineage 5 virus can increase its proportion of representation.

Experiments are logically progressed through and despite the mostly negative findings in that do not identify a mechanism behind the increased replication the manuscript is interesting and makes an important finding.

There are two main concerns that need to be addressed:

1.) the statistical analyses performed may be problematic (this is not my expertise, but it appears that there are potentially some issues here). There are very small differences between a lot of the data (figure 2 for example), yet the results are highly significant. It also seems very surprising that these small difference would hold out to be biologically significant. For example, the data in most of figure 2, but especially 2c despite being only marginally different (less than 0.2log in some cases), is highly significant ($p < 0.0002$). Figure 2A also seems unlikely to be significant.

Another example is figure 6 where despite there being various difference between the data points and SD bars all samples have the same significant of < 0.0332 . And again the differences observed are very small.

Again in Figure 9 there are two very specific P values

And in figure 10 a Mann-Whitney is used to analyze data through time with a single p value being given. But then for the the lung explant data a Krustall-Wallis test is used (which seems more appropriate and should have been used for A).

It seems that t-test or Mann Whitney tests were used for the majority of the data analysis. These are suitable to compare paired samples. But in most cases it would seem that multiple comparison would be necessary and simply repeating theses test multiple times introduces error that may have not been accounted for.

2.) if lineage 5 viruses have better replication capacity and the sequences of the viruses used in this study are known, why is there no attempt to link amino acid (or nucleotide) changes to this finding? It is unclear if this was just not done, or if it was but did not yield a clear answer. This should be addressed somehow.

Minor comments:

In49 - "swept other" is strange wording, the meaning is understood, but "replaced" might be more understandable

In178 - analyzing the peak heights of Sanger sequencing is not really the best method to quantify SNPs; however, the data seems consistent and is not overstated. There are more quantitative methods to achieve this.

In313 - what change in phenotype does lineage 5 bring to humans, please explain

In320 - what evidence is there that lineage 5 has increased transmissibility or virulence, please include; the finding here is that binding is the same. Virulence was not assessed. also how can you be certain that increased replication in a cell line leads to higher excreted virus

In321 - assuming higher doses are transmitted because of increased replication (not necessarily) how does this lead to a higher capacity to adapt to humans?

In324 - provide references that indicate that decreased cytokines is linked to increased virulence. In reservoir where they may be extensive virus replication, there is usually limited immune activation and little to no pathology.

Fig10 - the number of fields looked at for the analysis should be indicated

Reviewer #2 (Remarks to the Author):

The provided manuscript provided interesting study about the replication potential of lineage 5 MERS CoV isolates by using various techniques. However, several major concerns have to be covered.

- At first, the title, which comprises the word "increased zoonotic potential" is not supported by proper experimental evidences. The content of manuscript is rather describes the infectious capacity in cell cultures and lung ex-vivo transplants. There was no any studies to support the concept of zoonosis or increased zoonotic potential. Therefore, the title must be modified to imply the investigations of the lineage 5 isolates replication potency in vitro and in cell cultures.
- The viruses used in the study were stock viruses from Saudi Arabian patients diagnosed with MERS between March 2014 and November 2015. The obtained data cannot be conveyed to the circulating viruses at the moment. The data must be reanalyzed in the light of the recent changes in virus genome sequences and to confirm that lineage 5 are still circulating or there might be changes in the genome composition, phylogenetics and infectivity. The samples were taken about 5-6 years ago, its application to the recent status of circulating viruses is questionable.
- The isolates of this study are from two towns in Saudi Arabia, Jeddah and Riyadh. The lack of diverse samples or covering different geographic distribution underscores the driven conclusions about the nature of circulating viruses.
- In lines 89-91, and the reference #18, samples were taken from imported and local camels in Saudi Arabia between 2016 and 2018. Based on these studies the current manuscript stated that "Surveillance of circulating viruses has been ineffective after 2015". This seems to be based on extreme bias in sampling. Two recombination events were predicted in Jeddah and Riyadh. The subsequent sampling and the provided sampling in this manuscript were constantly from Jeddah and Riyadh. The provided conclusion must be based on samples from different geographic areas and up dated to the recent isolates.
- All isolates seem to be from human sources. The history associated with these samples was not provided. The role of camels or zoonotic aspects and its relation to the observed virus characteristics was not justified.
- Figure 1 includes data from sequences as low as 5000 nucleotides. How these short sequences were related to the about >29000 nucleotides genomes?. The highly diverse sequences lengths

cannot be used to deliver a conclusion.

- According to Fig.1, the number of sequences in 2016-2019 is very low compared with 2013-2015. The epidemiological data of these isolates were not provided. Given the lack of diversity and geographic coverage of sampling, the recent isolates might be originating in a single place or two towns with a single virus lineage or a virus circulating in a single hospital intensive care facility.
- According to Fig. 2, lineage 5 showed enhanced replication in Calu-3 cells. Therefore, the result from the direct competition assay might be biased toward lineage 5 isolates. This test could be repeated by using other reproducible assay or repeated in Vero cells, which showed equal response to the different virus lineages.
- Despite the discovery of the lineage 5 several years ago, no study provided the molecular mechanisms underlying the circulation of this virus and the enhanced cell infection in this manuscript. The results of INF sensitivity is still controversial. The authors must provide virus specific factors associated with the observed enhanced infectivity. For instance, viral PLpro and UBL are well known for their interference with their host natural immune responses. What is the status of these proteins in lineage 5?
- A last important question, as described, lineage 5 is the circulating form after 2015. Despite the detected higher infectivity in cells and ex-vivo lung transplants, the announced number of infected cases by the Saudi Ministry of Health reports are much lower than the recorded numbers in the era of other lineages. What is the explanation of this notice?

Reviewer #3 (Remarks to the Author):

General comments

Based on the analysis of MERS-CoV strain sequences circulating in Saudi Arabia between 2013 and 2019, the authors have first concluded that a novel recombinant clade (MERS-CoV lineage 5) that caused nosocomial outbreaks in Arabia Saudi and South Korea emerged, overgrowing other circulating clades such as parental lineages 3 and 4. The new selected lineage 5 was dominant in year 2015 and almost the only one circulating from 2016 up to 2019, when this part of the study ended. Since then, lineage 5 remains the only one circulating in Saudi Arabia. In an attempt to identify the molecular basis for the selection of lineage 5, authors have performed a comprehensive comparative analysis of representative members of lineages 3, 4 and 5, including replication levels, competitive replicative fitness, sequence of the receptor binding site of each virus, cell entry capacity and receptor binding avidity, virus neutralization by MERS-CoV specific human and camel sera, differential cytokine induction, IFN sensitivity, JAK/STAT signaling pathway sensitivity to IFN treatment, and replication phenotype in models of the human respiratory tract.

The main conclusion of this work has been that the newly emerging genetic lineage of MERS-CoV has higher replicative fitness and causes a lower level of cytokine induction than previously circulating strains, and have concluded that the emergence and imposition of lineage 5 was due to the recombination of previously circulating strains, leading to lineage 5 circulating both in camels and humans. These data suggested that the recombinant virus was originated in camels and then transferred to humans. Lineage 5 isolates reached average titers up to 15-fold higher compared to lineages 3, lineage 4 and the EMC reference MERS-CoV reference strain. The present study provides the first comprehensive phenotypical assessment of lineage 5 isolates, showing changes that likely correlate with transmissibility and virulence. This is a strait forward study and I consider that this manuscript has novel information of high interest to understand the evolution and dissemination of MERS-CoV in the World.

Specific comments:

1. Lineage 5 virus has clearly shown that circulates in humans, the origin of the sequences used in this study. Have authors address the question on whether this recombinant virus was already circulating in camels before crossing species barrier to humans, by also analyzing virus sequences of virus circulating in camels during 2015?
2. Lines 254-257. It is indicated that "the two lineages 5 isolates used in the experiment replicated

to a higher level than the members of lineage 3 and 4 (two strains each). To our surprise, lineage 5 induced significantly lower levels of IFN and CCL5 mRNA compared to isolates pertaining to lineage 3 and 4" In contrast, this reviewer would expect to see this result, in fact, lower IFN levels should help higher virus titers. Please, explain your surprise. Furthermore, the average levels of lineage 5 replication are 15-fold higher than those of lineages 3 or 4, justifying the fast and clear selection of lineage 5 viruses over those of lineages 3 and 4 (lines 293-294).

3. Lines 352-354. Discussion, extension of point 1. It was considered whether recombinant virus selection could have taken place within the animal reservoir. Was lineage 5 already present in camels? Please, provide direct evidence supporting this statement. Do lineage 5 viruses also replicate more efficiently than lineages 3 and 4 in camel cells.

4. Figure 1. The limitation on available MERS-CoV sequences in the databases may introduce a bias in the circulating lineages analysis. No doubt that lineage 5/NRC has been imposed over the other viruses in 2015. Nevertheless, considering that the total number of genomes analyzed from 2016 to 2019 range between 4.5% to 13% of those analyzed in 2015, it cannot be excluded that other lineages are still circulating since 2016. Please, clarify.

Minor comments:

1. Line 415. Why Vero E6 cells, and not Vero B4, as the cell line used in the other parts of the manuscript were used in this case? Is it a mistake, or because these experiments were performed in an alternative laboratory?

REPLY TO THE REVIEWER COMMENTS:

Reviewer #1 (Remarks to the Author):

A comparison of various isolates belonging to different lineages of MERS-CoV is performed in a number of cell lines and identifies that virus isolates from lineage 5 appear to have a replicative advantage over EMC2012 and other lineage viruses. Interestingly, it appears that lineage 5, which also happens to be a recombinant virus, has taken over to become the predominant lineage in Saudi Arabia. Based on limited data provided it appears that it rapidly displaced all other lineages and now is the only lineage circulating. However, this finding is underpowered by a lack of samples from years 2016 onwards - somewhat limiting this conclusion, but that is not a major flaw. Attempts are made to identify a mechanism behind this fitness gain through competition assays, entry assays (various), binding assays, and various replication (genome and virus) assays (including in primary cells). None of these mechanisms appear to be the cause of this finding. The authors do however identify that lineage 5 viruses induce a lesser innate immune response despite increased (or because of) increased replication and that lineage 5 viruses are less sensitive to IFN-induced virus replication restriction.

The increased fitness effect is not that prevalent in most of the studies however it is consistent and possibly the strongest evidence is that virus competition assays where even at a ratio of 9:1, the lineage 5 virus can increase its proportion of representation.

Experiments are logically progressed through and despite the mostly negative findings in that do not identify a mechanism behind the increased replication the manuscript is interesting and makes an important finding.

There are two main concerns that need to be addressed:

1.) the statistical analyses performed may be problematic (this is not my expertise, but it appears that there are potentially some issues here). There are very small differences between a lot of the data (figure 2 for example), yet the results are highly significant. It also seems very surprising that these small difference would hold out to be biologically significant. For example, the data in most of figure 2, but especially 2c despite being only marginally different (less than 0.2log in some cases), is highly significant ($p < 0.0002$). Figure 2A also seems unlikely to be significant.

Another example is figure 6 where despite there being various difference between the data points and SD bars all samples have the same significant of < 0.0332 . And again the differences observed are very small.

Again in Figure 9 there are two very specific P values

And in figure 10 a Mann-Whitney is used to analyze data through time with a single p value being given. But then for the the lung explant data a Krustall-Wallis test is used (which seems more appropriate and should have been used for A).

It seems that t-test or Mann Whitney tests were used for the majority of the data analysis. These are suitable to compare paired samples. But in most cases it would seem that multiple comparison would be necessary and simply repeating theses test multiple times introduces error that may have not been accounted for.

R: We have consulted with a statistician and adapted the statistics and display of significance results throughout the manuscript. All group-wise comparisons for single time point experiments (Figure 2, Figure 7, Figure 8, Figure 9 and Figure 10) are now consequently done by two-tailed Student's t-tests. For paired data sets (time course experiments, Figure 6 and Figure 10), we now applied an ANOVA, followed by single t-tests between data points of the same time point (Figure 10).

2.) if lineage 5 viruses have better replication capacity and the sequences of the viruses used in this study are known, why is there no attempt to link amino acid (or nucleotide) changes to

this finding? It is unclear if this was just not done, or if it was but did not yield a clear answer. This should be addressed somehow.

R: We agree. In the revised manuscript we provide information on amino acid differences between the viral lineages in **Supplementary table 3** (supplements, pp. 10-11). We now specify which of these mutations could contribute to changes in phenotype, both in the **results** (line 307-326) and in the **discussion** (line 387-409).

Because of the absence of relevant changes in spike, we focused our search for potentially relevant mutations on viral interferon antagonists. There were no mutations unique and specific for all lineage 5 viruses in functional domains of previously described interferon antagonists, except for an M6T mutation in protein 4b. To address the influence of this mutation on viral replication, we constructed a recombinant MERS-CoV with the p4b M6T mutation. We did not observe changes in replication driven by this mutation (now shown in **Supplementary Figure 5** (supplements, page 6). These aspects are all covered in the revised version of the paper including the discussion:

(excerpt):

“We particularly focused on amino acid positions in viral proteins for which structural data are available (PLPro in nsp3) and on proteins uniquely attributed to interferon antagonism in MERS-CoV (p4a and p4b). The PLPro in nsp3 of most CoVs encodes for a deubiquitinating domain, which seems to counteract the antiviral function of ISG15, and importantly, the transcriptional activity of IRF3, antagonizing IRF3-mediated induction of IFNs and cytokines downstream of virus sensing (49, 50, 56-58). A second motif in nsp3, the macro domain, may further contribute to nsp3-mediated immune antagonism in SARS- and related CoVs by its ADP-ribose-1'-phosphatase domain that seems to decrease IFN and ISG induction in infected cells by a yet unknown mechanism (38). However, all divergent amino acids lay outside of the macro domain and described functional residues of PLPro (37, 59). MERS-CoV accessory protein 4a antagonizes IFN induction by binding to dsRNA and inhibiting MDA5 activation, possibly by binding to its activator PACT (29, 60). However, no lineage specific amino acid exchanges are present in p4a of lineage 5 MERS-CoV. There was one potentially relevant exchange in p4b, a functional antagonist of RNase L. We tested the effect of this exchange by introducing it into a MERS-CoV reference strain via reverse genetics (41). However, the change did not influence replication level.”

There is also a discussion of limitations and future perspectives toward a mechanistic understanding of fitness changes, which we do not copy here for reasons of space.

Minor comments:

In49 - "swept other" is strange wording, the meaning is understood, but "replaced" might be more understandable

R: We agree and have changed the phrasing accordingly. Line 98: “swept” was changed to “replaced”.

In178 - analyzing the peak heights of Sanger sequencing is not really the best method to quantify SNPs; however, the data seems consistent and is not overstated. There are more quantitative methods to achieve this.

R: We have indeed done a part of the population sequencing by NGS, but saw no analytical advantage for our purpose and then switched back so Sanger sequencing. NGS is capable of detecting smaller background populations but these are irrelevant for the present experiments as both populations remain detectable by our approach over the entire experiment. Also, a very exact quantification is not necessary as we follow the population composition over several passages and clearly show that trends are

consistent with little variation from passage to passage while showing the overall trend caused by relative fitness. It is more important to follow relative fitness over serial passages than to determine population fractions with utmost analytical precision (e.g., by using NGS instead of Sanger). Many studies analyze only one cycle of population expansion which may be considered insufficient to really confirm relative fitness.

In313 - what change in phenotype does lineage 5 bring to humans, please explain

And

In320 - what evidence is there that lineage 5 has increased transmissibility or virulence, please include; the finding here is that binding is the same. Virulence was not assessed. also how can you be certain that increased replication in a cell line leads to higher excreted virus

And

In321 - assuming higher doses are transmitted because of increased replication (not necessarily) how does this lead to a higher capacity to adapt to humans?

R: We answer these points collectively as they are related.

These aspects are now covered in the discussion, line 367-375: "The *in-vitro* and *ex-vivo* cell culture models used in our study cannot definitively reflect virus excretion *in-vivo*, but the observed increase in fitness in these models agrees with epidemiological trends over several years, in a large geographic region. Even slight increases in replication may have profound long-term influence on pathogen fitness in the host population. As the increase in replication was also observed in human-derived systems, increased virus shedding by lineage 5 in humans is conceivable. Because the transmitted virus dose is positively correlated with the rate of adaptive evolution, lineage 5 may pose an increased risk of human adaptation when transmission chains occur in nosocomial outbreaks (45, 46)."

We acknowledge reviewer 1's remark that assessments of virus virulence cannot be gained in cell culture models. Even though one could argue that the identified difference in cytokine induction is a virulence trait, we accede to the reviewer's recommendation and deleted statements about virulence. We are now mainly referring to the observed effects as differences in replication, potential shedding, and fitness.

In324 - provide references that indicate that decreased cytokines is linked to increased virulence. In reservoir where they may be extensive virus replication, there is usually limited immune activation and little to no pathology.

R: This is also covered in our reply above. There are many instances in which attenuation in live vaccine candidate strains can be linked to increased cytokine induction (e.g., Rift Valley Fever Virus clone 13, Influenza A delta NS1, etc.). However, as we have removed claims regarding virulence, this is of no matter here.

Fig10 - the number of fields looked at for the analysis should be indicated

R: All lung explant pieces were cut into defined sizes and the complete piece of lung explant was analyzed for each sample. We state this in the methods section and included a statement in the Figure legend (line 834-838).

Reviewer #2 (Remarks to the Author):

The provided manuscript provided interesting study about the replication potential of lineage 5 MERS CoV isolates by using various techniques. However, several major concerns have to be covered.

- At first, the title, which comprises the word “increased zoonotic potential” is not supported by proper experimental evidences. The content of manuscript is rather describes the infectious capacity in cell cultures and lung ex-vivo transplants. There was no any studies to support the concept of zoonosis or increased zoonotic potential. Therefore, the title must be modified to imply the investigations of the lineage 5 isolates replication potency in vitro and in cell cultures.

R: In the revised version of the manuscript we now performed virus infection assays in camel primary lung epithelium and confirmed increased replication of lineage 5 MERS-CoV isolates (**new Figure 10C**). We feel these findings support the idea that lineage 5 MERS-CoV might also have increased zoonotic potential. However, we agree with reviewer 2 that the concept of zoonotic transmission is not the main focus of the manuscript. We therefore modified the title and changed “zoonotic potential” to “replicative fitness”.

- The viruses used in the study were stock viruses from Saudi Arabian patients diagnosed with MERS between March 2014 and November 2015. The obtained data cannot be conveyed to the circulating viruses at the moment. The data must be reanalyzed in the light of the recent changes in virus genome sequences and to confirm that lineage 5 are still circulating or there might be changes in the genome composition, phylogenetics and infectivity. The samples were taken about 5-6 years ago, its application to the recent status of circulating viruses is questionable.

R: In order to address this limitation we included a MERS-CoV strain isolated in 2017 in Dubai that also belongs to lineage 5 but is phylogenetically distinct from the isolates used previously (new phylogenetic tree in **Supplementary Figure 1**, supplements, page 1). We confirm that the MERS-CoV 2017 strain has the same replicative fitness as other lineage 5 isolates indicating that the replication phenotype of lineage 5 was not lost in MERS-CoV strains isolated later than 2015 (**new supplementary Figure 2**, supplements, page 3). In addition, while this manuscript was under revision, an epidemiological study was published that shows that lineage 5 remains the dominant lineage in circulation (Hemida et al., EID, 2020). This is referenced and discussed in the revised manuscript.

- The isolates of this study are from two towns in Saudi Arabia, Jeddah and Riyadh. The lack of diverse samples or covering different geographic distribution underscores the driven conclusions about the nature of circulating viruses.

and

- In lines 89-91, and the reference #18, samples were taken from imported and local camels in Saudi Arabia between 2016 and 2018. Based on these studies the current manuscript stated that “Surveillance of circulating viruses has been ineffective after 2015”. This seems to be based on extreme bias in sampling. Two recombination events were predicted in Jeddah and Riyadh. The subsequent sampling and the provided sampling in this manuscript were constantly from Jeddah and Riyadh. The provided conclusion must be based on samples from different geographic areas and up dated to the recent isolates.

R: Collective answer to two comments: To further extend the temporal and geographical range of isolates under study, we included a MERS-CoV lineage 5

isolate from Dubai sampled in 2017 as mentioned above. The nomenclature of the strains “Jeddah” and “Riyadh” refers to the respective regional labs that we obtained the samples from. Both regional labs collect samples from hospitals all over the country, however, the Jeddah lab with a focus on the East coast region. As all samples were fully anonymized, we cannot provide a detailed history for each individual sample. The different phylogenetic clustering indicates that samples were retrieved from multiple sites. We have modified several paragraphs to better indicate the origin of human samples used for this study (results, line 131-140).

- All isolates seem to be from human sources. The history associated with these samples was not provided. The role of camels or zoonotic aspects and its relation to the observed virus characteristics was not justified.

R: Phylogenetic analysis of MERS-CoV strains from the Arabian Peninsula show that there are no major differences between dromedary camel and human MERS-CoV isolates (Dudas, eLife, 2018, new reference 60), i.e., samples taken from humans provide a sample of the diversity of circulating virus populations in dromedary camels. We have included a selection of references for this (ref 14-17, new reference 60). Regarding the concept of zoonotic transmission, we agree that this was not a major focus of our study. We therefore changed the title (“zoonotic potential” was changed to “replicative fitness”) and modified several paragraphs to better indicate the origin of human samples used for this study (results, line 131-140).

- Figure 1 includes data from sequences as low as 5000 nucleotides. How these short sequences were related to the about >29000 nucleotides genomes? The highly diverse sequences lengths cannot be used to deliver a conclusion.

R: As pointed out in our previous study (see SA El-Kafrawy, Lancet Planetary Health, 2019), we used concatenated sequences that provide robust resolution in phylogenetic analyses. We clarified this in the figure legend.

- According to Fig.1, the number of sequences in 2016-2019 is very low compared with 2013-2015. The epidemiological data of these isolates were not provided. Given the lack of diversity and geographic coverage of sampling, the recent isolates might be originating in a single place or two towns with a single virus lineage or a virus circulating in a single hospital intensive care facility.

R: As mentioned above the sample codes are based on the two different regional labs that provided the samples. As seen by the differential phylogenetic clustering the samples are from multiple locations all around Jeddah and Riyadh region. We have modified several paragraphs to better indicate the origin of human samples used for this study (results, line 131-140).

- According to Fig. 2, lineage 5 showed enhanced replication in Calu-3 cells. Therefore, the result from the direct competition assay might be biased toward lineage 5 isolates. This test could be repeated by using other reproducible assay or repeated in Vero cells, which showed equal response to the different virus lineages.

R: The lineage 5 MERS-CoV phenotype was confirmed in multiple cell lines and cell culture systems including primary lung epithelial cells from humans and now camels and in human *ex-vivo* lung explants. The applied Vero cells would not be well suited for competition assays, as we provide evidence that the lineage 5 phenotype might correlate with innate immune signaling, which is impaired in Vero cells (lack of type I IFN loci).

- Despite the discovery of the lineage 5 several years ago, no study provided the molecular mechanisms underlying the circulation of this virus and the enhanced cell infection in this manuscript. The results of INF sensitivity is still controversial. The authors must provide virus specific factors associated with the observed enhanced infectivity. For instance, viral PLpro and UBL are well known for their interference with their host natural immune responses. What is the status of these proteins in lineage 5?

R: We addressed this in detail in our reply to reviewer 1. We fully agree with the reviewer that there is interest in mechanisms. The mentioned PLpro/UBL activities are among the known mechanisms by which the virus interacts with cellular immune functions. These and other known interactions have become part of our efforts to provide better mechanistic insight.

1. Amino acid changes between the MERS-CoV lineages have been identified and are now listed in the **new Supplementary Table 3** (supplements, pp. 10-11).
2. We constructed a recombinant MERS-CoV that harbours a potentially relevant M6T exchange in p4b as per our assessment of amino acid exchanges and studied the replication phenotype of this variant (**new Supplementary Figure 5**, supplements, page 7).
3. We revised the manuscript in compliance with reviewer 3's specific suggestions related to amino acid changes in PLpro and UBL (both in nsp3). We have added paragraphs to the **results** (line 307-326) and the **discussion** (line 387-409) that summarize our amino acid sequence analyses of all viral proteins for the viruses included in the study. In these, we particularly focused on a possible role of nsp3 and other described interferon antagonists. We have referenced all publications that were used for sequence comparisons in functional protein domains.

- A last important question, as described, lineage 5 is the circulating form after 2015. Despite the detected higher infectivity in cells and ex-vivo lung transplants, the announced number of infected cases by the Saudi Ministry of Health reports are much lower than the recorded numbers in the era of other lineages. What is the explanation of this notice?

R: We had previously briefly touched on this in the introduction ("Surveillance of circulating viruses has been ineffective after 2015.") without giving further details. In the revised version we added a paragraph and a reference to summarize the proposed reasons for a reduction in notified MERS cases, introduction, line 95-98 "Limited diagnostic testing, limited surveillance and improved hospital infection control may have caused an apparent decline of notified cases after 2015 (18). Only based on more recent studies, it appears that lineage 5 has essentially replaced all other endemic strains since 2015 (19, 20) (**Figure 1**)."

(new reference 18: Donnelly, EID, 2019).

We also want to point out that a further epidemiological study was published while we revised this manuscript. This study also finds lineage 5 MERS-CoV to be the dominantly circulating lineage in Saudi Arabian camels (new reference 20: Hemida et al., EID, 2020).

Reviewer #3 (Remarks to the Author):

General comments

Based on the analysis of MERS-CoV strain sequences circulating in Saudi Arabia between 2013 and 2019, the authors have first concluded that a novel recombinant clade (MERS-CoV lineage 5) that caused nosocomial outbreaks in Arabia Saudi and South Korea emerged, overgrowing other circulating clades such as parental lineages 3 and 4. The new selected lineage 5 was dominant in year 2015 and almost the only one circulating from 2016 up to 2019, when this part of the study ended. Since then, lineage 5 remains the only one circulating in Saudi Arabia. In an attempt to identify the molecular basis for the selection of lineage 5, authors have performed a comprehensive comparative analysis of representative members of lineages 3, 4 and 5, including replication levels, competitive replicative fitness, sequence of the receptor binding site of each virus, cell entry capacity and receptor binding avidity, virus neutralization by MERS-CoV specific human and camel sera, differential cytokine induction, IFN sensitivity, JAK/STAT signaling pathway sensitivity to IFN treatment, and replication phenotype in models of the human respiratory tract.

The main conclusion of this work has been that the newly emerging genetic lineage of MERS-CoV has higher replicative fitness and causes a lower level of cytokine induction than previously circulating strains, and have concluded that the emergence and imposition of lineage 5 was due to the recombination of previously circulating strains, leading to lineage 5 circulating both in camels and humans. These data suggested that the recombinant virus was originated in camels and then transferred to humans. Lineage 5 isolates reached average titers up to 15-fold higher compared to lineages 3, lineage 4 and the EMC reference MERS-CoV reference strain. The present study provides the first comprehensive phenotypical assessment of lineage 5 isolates, showing changes that likely correlate with transmissibility and virulence. This is a strait forward study and I consider that this manuscript has novel information of high interest to understand the evolution and dissemination of MERS-CoV in the World.

Specific comments:

1. Lineage 5 virus has clearly shown that circulates in humans, the origin of the sequences used in this study. Have authors address the question on whether this recombinant virus was already circulating in camels before crossing species barrier to humans, by also analyzing virus sequences of virus circulating in camels during 2015?

R: Yes, the recombinant lineage was already circulating in camels before (reference 14-17 in the manuscript). We addressed this question already in a previous publication (El-Kafary, Lancet Planetary health, cited) and confirmed the existence of lineage 5 MERS-CoV in dromedaries on the Arabian Peninsula.

2. Lines 254-257. It is indicated that “the two lineages 5 isolates used in the experiment replicated to a higher level than the members of lineage 3 and 4 (two strains each). To our surprise, lineage 5 induced significantly lower levels of IFN and CCL5 mRNA compared to isolates pertaining to lineage 3 and 4” In contrast, this reviewer would expect to see this result, in fact, lower IFN levels should help higher virus titers. Please, explain your surprise. Furthermore, the average levels of lineage 5 replication are 15-fold higher than those of lineages 3 or 4, justifying the fast and clear selection of lineage 5 viruses over those of lineages 3 and 4 (lines 293-294).

R: We were intrigued by this phenotype as one might have hypothesized that increased replication leads to an increased accumulation of viral RNA, triggering a stronger innate immune response. The mechanisms of immune induction and evasion

should be quite similar among so closely related virus strains. This leads us to suspect that the observed phenotypic differentiation is linked to differences in the avoidance of cytokine induction.

In the revised manuscript, we have more clearly linked the concept of pattern recognition to descriptions of our initial observations (results, line 263-277): “To explore if higher replication is accompanied by higher cytokine induction, we infected Calu-3 cells with two viruses of each phylogenetic lineage in a single-cycle infection for 12 hours and analyzed the mRNA expression levels of a set of immune-related genes. We chose IRF3-regulated genes IFN β 1 and IFNL1, NF κ B-regulated genes CCL5 and TNF α , as well as the IFN-stimulated gene (ISG) Mx1. To enable single cycle virus infection, cells were infected at MOI = 2 (**Figure 7A**). Also under these conditions, the two lineage 5 isolates used in the experiment replicated to a higher level than the members of lineage 3 and 4. Despite higher replication putatively upregulating ligands for innate immune sensing such as dsRNA, lineage 5 induced significantly lower levels of IFN and CCL5 mRNA compared to isolates pertaining to lineage 3 and 4 (**Figure 7B**). Reduced IFN β 1 and IFNL1 mRNA expression levels in lineage 5 infected cells might be a result of a more effective viral nucleic acid sequestration from pattern recognition receptors (PRRs) recognition or may reflect an active block of the innate immune activation by viral antagonists (35). Immune gene mRNA induction in general seemed to be highest with lineage 3 strains.”

3. Lines 352-354. Discussion, extension of point 1. It was considered whether recombinant virus selection could have taken place within the animal reservoir. Was lineage 5 already present in camels? Please, provide direct evidence supporting this statement. Do lineage 5 viruses also replicate more efficiently than lineages 3 and 4 in camel cells.

R: This is covered in our reply to question 1. We referenced all papers that describe the recombination event leading to the formation of lineage 5 (reference 14-17). The revised manuscript now includes virus replication data showing that lineage 5 infection phenotype is also observed in primary camel lung epithelium cells, **new Figure 10C**.

4. Figure 1. The limitation on available MERS-CoV sequences in the databases may introduce a bias in the circulating lineages analysis. No doubt that lineage 5/NRC has been imposed over the other viruses in 2015. Nevertheless, considering that the total number of genomes analyzed from 2016 to 2019 range between 4.5% to 13% of those analyzed in 2015, it cannot be excluded that other lineages are still circulating since 2016. Please, clarify.

R: In principle, we agree with the reviewer that the limited availability of sequences from 2016-2019 might bias the analyses. Therefore, we phrased the matter carefully, line 96-100: “Limited diagnostic testing, limited surveillance and improved hospital infection control may have caused an apparent decline of notified cases after 2015 (18). Only based on more recent studies, it appears that lineage 5 has essentially replaced all other endemic strains since 2015 (19, 20) (**Figure 1**).”

We also want to point out that a further epidemiological study was published while we revised our manuscript. This study confirms that lineage 5 MERS-CoV are the only remaining lineage circulating in Saudi Arabian camels (new reference 20: Hemida et al., EID, 2020).

Minor comments:

1. Line 415. Why Vero E6 cells, and not Vero B4, as the cell line used in the other parts of the manuscript were used in this case? Is it a mistake, or because these experiments were performed in an alternative laboratory?

R: This is not a mistake. The human lung explant infection was carried out by a collaborating group that used Vero E6 for titration of virus. The results are equivalent and the use of another subtype of the Vero cell line does not influence the validity or result of the experiment.

Reviewer comments, second round –

Reviewer #2 (Remarks to the Author):

All comments were properly covered in the revised version.

Reviewer #3 (Remarks to the Author):

Review on the manuscript by

I consider that the reviewer has either properly replied our questions or modified the content of the new version of the submitted paper to properly adjust their statements by rewriting the text and providing a correct conclusion. In fact, one of the main outcomes of the manuscript, stated in the title of the paper, has been tuned down by replacing references related to increase in virulence by other referring to "increased replicative fitness of the recombinant lineage 5"

Also, in reply to our question authors have also clarified that the two lineage 5 isolates used in the experiments replicated to a higher level than the members of lineage 3 and 4. Despite higher replication putatively upregulating ligands for innate immune sensing such as dsRNA, lineage 5 induced significantly lower levels of IFN and CCL5 mRNA compared to lineage 3 and 4 isolates.

Authors have identified that MERS-CoV lineage 5 has displaced all other lineages and now is the only one circulating at least up to 2019. Furthermore, this conclusion has already been supported during the revision of this manuscript by other publications that have reached the same conclusion during the last months (as an example, the publication quoted in new reference 20). In addition, although the authors have not provided a detailed mechanism responsible for this dominance of lineage 5, have clearly shown that this lineage induces a lesser innate immune response despite its increased replication, and that lineage 5 viruses are less sensitive to IFN-induced virus replication restriction.

In addition, in reply to our questions the authors have incorporated novel information (Fig. 10C) confirming that lineage 5 circulates both in humans and also in camels and that, in fact it was already circulating in camels before than in humans.

Overall, I consider that the new version of the manuscript has properly addressed the points raised by the reviewers and represents a major contribution of interest to the scientific community.

Response to REVIEWERS' COMMENTS

Reviewer #2 (Remarks to the Author):

All comments were properly covered in the revised version.

R: We thank the reviewer #2 for reviewing our work and for making suggestions on how to improve the manuscript.

Reviewer #3 (Remarks to the Author):

Review on the manuscript by

I consider that the reviewer has either properly replied our questions or modified the content of the new version of the submitted paper to properly adjust their statements by rewriting the text and providing a correct conclusion. In fact, one of the main outcomes of the manuscript, stated in the title of the paper, has been tuned down by replacing references related to increase in virulence by other referring to "increased replicative fitness of the recombinant lineage 5"

Also, in reply to our question authors have also clarified that the two lineage 5 isolates used in the experiments replicated to a higher level than the members of lineage 3 and 4. Despite higher replication putatively upregulating ligands for innate immune sensing such as dsRNA, lineage 5 induced significantly lower levels of IFN and CCL5 mRNA compared to lineage 3 and 4 isolates.

Authors have identified that MERS-CoV lineage 5 has displaced all other lineages and now is the only one circulating at least up to 2019. Furthermore, this conclusion has already been supported during the revision of this manuscript by other publications that have reached the same conclusion during the last months (as an example, the publication quoted in new reference 20). In addition, although the authors have not provided a detailed mechanism responsible for this dominance of lineage 5, have clearly shown that this lineage induces a lesser innate immune response despite its increased replication, and that lineage 5 viruses are less sensitive to IFN-induced virus replication restriction.

In addition, in reply to our questions the authors have incorporated novel information (Fig. 10C) confirming that lineage 5 circulates both in humans and also in camels and that, in fact it was already circulating in camels before than in humans.

Overall, I consider that the new version of the manuscript has properly addressed the points raised by the reviewers and represents a major contribution of interest to the scientific community.

R: R: We thank the reviewer #3 for reviewing our work and for making suggestions on how to improve the manuscript.